# Amplification of light absorption of black carbon associated with air pollution

Yuxuan Zhang[1,2], Qiang Zhang[1], Yafang Cheng[3,2], Hang Su[3,2], Haiyan Li[4], Meng Li[1,2], Xin Zhang[1], Aijun Ding[5], and Kebin He[4]

[1]Ministry of Education Key Laboratory for Earth System Modeling, Department of Earth System Science, Tsinghua University, Beijing 100084, China

[2]Multiphase Chemistry Department, Max Planck Institute for Chemistry, Mainz 55020, Germany

[3]Institute for Environmental and Climate Research, Jinan University, Guangzhou 510630, China

[4]State Key Joint Laboratory of Environment Simulation and Pollution Control, School of Environment, Tsinghua University, Beijing 100084, China

[5]Institute for Climate and Global Change Research, School of Atmospheric Sciences, Nanjing University, Nanjing, China

*Correspondence to*: Qiang Zhang (qiangzhang@tsinghua.edu.cn)

**Abstract.** The impacts of black carbon (BC) aerosols on air quality, boundary layer dynamic and climate depend not only on the BC mass concentration but also on the light absorption capability of BC. It is well known that the light absorption capability of BC depends on the amount of coating materials (namely other species on BC by condensation and coagulation). However, the difference of light absorption capability of ambient BC-containing particles under different air pollution conditions (e.g., the air clean and polluted conditions) remains unclear due to the complex aging process of BC in the atmosphere. In this work, we investigated the evolution of light absorption capability for BC-containing particles with changing pollution levels in urban Beijing, China. During the campaign period (17 to 30 November 2014), with the increase of $PM_1$ concentration from ~10 μg $m^{-3}$ to ~230 μg $m^{-3}$, we found that the mass-weighted averages of the aging degree and theoretical light absorption capability of BC-containing particles increased by ~33% and ~18%, respectively, indicating stronger light absorption capability of BC-containing particles under more polluted conditions due to more coating materials on the BC surface. By using effective emission intensity (EEI) model, we further found that aging during the regional transport plays an important role in the difference among the light absorption capability of BC-containing particles under different air pollution levels. During the pollution episode, ~63% of the BC over Beijing originated from regional sources outside of Beijing. These regionally sourced BC-containing particles were characterized by more coating materials on BC surface due to more coating precursors within more polluted air, which contributed ~75% of the increase in theoretical light absorption capability of BC observed in Beijing during the polluted period ($PM_1$ of ~230 μg $m^{-3}$) comparing to that in the clean period ($PM_1$ of ~10 μg $m^{-3}$). Due to the increase

of theoretical light absorption capability of BC associated with air pollution, the direct radiative forcing of BC was estimated to be increased by ~18% based on a simple radiation transfer model. Our work identified an amplification of theoretical light absorption and direct radiative forcing under more air polluted environment due to more coating materials on BC. The air pollution control measures may, on the other hand, break the amplification effect by reducing emissions of both BC and the coating precursors and achieve co-benefits of both air quality and climate.

## 1 Introduction

Black carbon (BC) is an important aerosol component that absorbs visible sunlight and contributes to heating of the atmosphere (Bond and Bergstrom, 2006; Gustafsson and Ramanathan, 2016; Menon et al., 2002). Atmospheric BC can impact climate through radiative effects, which are strongly associated with the optical properties of BC, especially the light absorption (Cheng et al., 2006; Jacobson, 2000; Lesins et al., 2002; Ramanathan and Carmichael, 2008). Estimating the climate effects of BC is one of the major challenges in climate change research, partly due to large uncertainties in the light absorption capability of BC-containing particles under ambient conditions (Cappa et al. 2012; Liu et al. 2015; Liu et al. 2017; Gustafsson and Ramanathan, 2016). The light absorption capability of atmospheric BC is complex and poorly quantified, and it changes with the morphology, density and mixing state of the BC-containing particles (Knox et al., 2009; Peng et al. 2016; Schnaiter et al. 2005; Zhang et al., 2008). Previous theoretical (Jacobson, 2001; Moffet et al., 2009; Zhang et al., 2016) and observation (Cappa et al., 2012; Peng et al., 2016; Knox et al., 2009) studies showed a broad range of absorption enhancements (1.05-3.05) of BC-containing particles during the atmospheric aging process. To date, conflicts remain between model- and observation-based studies of the light absorption capability of atmospheric BC-containing particles (Cappa et al., 2012; Jacobson, 2001; Liu et al. 2015; Liu et al. 2017).

The light absorption capability of BC-containing particles depends strongly on the particle mixing state (Liu et al. 2015; Liu et al. 2017), i.e., the degree of internal mixing between BC and other particle species (i.e., non-BC components) by the atmospheric aging process (i.e., condensation, coagulation and heterogeneous oxidation). The non-BC species (i.e., coating materials) on the surface of BC cores can enhance BC light absorption via the lensing effect (namely, the coating materials act as a lens to focus more photons on BC, Bond et al., 2006; Fuller et al., 1999; Jacobson, 2001; Lack and Cappa, 2010). In terms of individual BC-containing particle, more coating materials result in its stronger light absorption capability. The coating materials on the BC surface are controlled by secondary processes (e.g., photochemical production) (Metcalf et al., 2013).

The production of secondary aerosols in the atmosphere varies significantly with pollution levels (Cheng, 2008; Mu et al., 2018; Zheng et al., 2016; Yang et al., 2015), indicating that BC-containing particles most likely exert different light absorption capability values under different pollution levels. Compared with air clean conditions, polluted periods feature more secondary aerosols, especially secondary inorganic species such as sulfate (Guo et al., 2014; Sun et al., 2014; Zheng et al., 2015). Whether the changes of secondary aerosols with air pollution will affect the coating materials on the BC is complex, which not only depends on the increase in BC amount versus secondary aerosols but also controlled by secondary material condensation on

BC versus non-BC containing particles. Recent BC aging measurements in Beijing and Houston using an environmental
chamber (flowing ambient air to feed with lab-generated fresh BC) have revealed that a clear distinction in the light absorption
capability of BC-containing particles exists between urban cities in developed and developing countries (Peng et al., 2016),
and this difference is likely due to the differences in air pollution levels.

5       To date, whether and how the aging degree and light absorption capability of BC-containing particles will change with air

pollution development is still unclear. Although the enhancement of BC light absorption due to coating materials on BC surface
has already been intensively investigated (Moffet et al., 2009; Schnaiter et al., 2005; Shiraiwa et al., 2010; Zhang et al., 2016),
there are few studies on the evolution of the light absorption capability of BC-containing particles with changing air pollution
levels. The variation in the light absorption capability of BC-containing particles associated with air pollution can lead to
different effects of BC aerosols on air quality and climate under different pollution levels. To improve the evaluation of BC-
related effects on air quality and climate, some models have considered BC internally mixed with other species (namely coating
materials on BC surface), which can affect the light absorption capability of BC-containing particles. However, the difference
of coating materials on BC under different air pollution conditions remains unclear.
In this work, we conducted an intensive field measurement campaign in urban Beijing, China, to investigate the difference
of the theoretical light absorption capability of atmospheric BC-containing particles under different pollution levels. Firstly,
we analyzed the evolution of theoretical light absorption capability of BC with increasing air pollution levels and estimated
the relationship between the changing rate in the theoretical light absorption capability of BC and that in the $PM_1$ or BC mass
concentrations. We then explored the cause of the evolution of theoretical light absorption capability of BC with increasing air
pollution levels and evaluated the relative importance of regional transport. Finally, we discussed the impact of changes in BC
light absorption capability with pollution levels on BC radiative forcing.
**2 Methods**
**2.1 Sampling site and measurements**
The *in-situ* measurements were conducted on the campus of Tsinghua University (Tsinghua site, 40º00'17" N, 116º19'34" E)
from November 17-30, 2014. The Tsinghua site (Fig. S1) is located in urban Beijing, China. The megacity Beijing is adjacent
to Hebei Province and the megacity Tianjin (Fig. S1), in which considerable industrial manufacturing has led to heavy
emissions of air pollution, especially in southern Hebei.
Ambient aerosol particles were collected by a $PM_1$ cyclone and then passed through a diffusion silica gel dryer, and they
were finally analyzed by an aethalometer (AE33, Magee Scientific Corp.), an Aerosol Chemical Speciation Monitor (ACSM,
Aerodyne Research Inc.) and a single particle soot photometer (SP2, Droplet Measurement Technologies Inc.). The AE33 can
measure the absorption coefficient ($\sigma_{ab}$) of sampled aerosols in seven spectral regions (370, 470, 520, 590, 660, 880 and 950
nm). At a wavelength of 880 nm, the absorption of aerosol particles is dominated by BC component because the light absorbed

by other aerosol components is significantly less (Drinovec et al., 2015; Sandradewi et al., 2008). In this study, the $\sigma_{ab}$ at 880 nm measured by the AE33 was used to characterize the theoretical light absorption of the BC-containing particles. More details on the AE33 measurement can be found in the work of Drinovec et al., 2015. Considering the filter-loading effect and multiple-scattering effect (Drinovec et al., 2015; Weingartner et al., 2003; Segura et al., 2014), the aethalometer data was corrected by compensation factors described in the supplementary information (Fig. S2 and the associated discussion). The ACSM and SP2 instruments measured the mass concentrations of non-refractory submicron-scale components (NR-PM1, i.e., sulfate, nitrate, ammonium, chloride and organics) and refractory BC (rBC), respectively, and the sum of these two measurements was used to estimate the $PM_1$ mass concentration. The ACSM instrument used in our study was described by Li et al. (2017).

The SP2 instrument measures a single BC-containing particle using a 1064 nm Nd:YAG intra-cavity laser beam. As the light-absorbing rBC passes through the laser beam and is heated to its vaporization temperature (~4000 K), it will emit incandescent light (i.e., visible thermal radiation), which is linearly proportional to the mass of the rBC (Metcalf et al., 2012; Moteki and Kondo, 2010; Schwarz et al., 2006; Sedlacek et al., 2012). In this study, the calibration curve of rBC mass vs. incandescence signal was obtained from the incandescence signal of size-resolved Aquadag particles (their effective density obtained from Gysel et al., 2011) using a DMA (differential mobility analyzers)-SP2 measurement system. Considering different sensitive of the SP2 to different rBC types (Gysel et al., 2011; Laborde et al., 2012), we corrected SP2 calibration curve by scaling the peak height of incandescence signal for Aquadag particles at each rBC mass based on the relationship between the sensitivity of SP2 to Aquadag and ambient rBC (Laborde et al., 2012). The particle-to-particle mass of ambient rBC can be determined by measuring its incandescence signal and comparing it to the calibration curve. The mass concentration of rBC is calculated from the particle-to-particle mass of rBC and the sampled flow (~0.12 lpm). Note that the SP2 detection efficiency (Fig. S3) have been considered in the calculation of rBC mass concentration. Furthermore, the scattering cross section of a BC-containing particle is obtained from its scattering signal using the leading edge only (LEO)-fit method (Gao et al. 2007). Zhang et al. (2016) has demonstrated the validity of the LEO-fit method for ambient BC-containing particles in China.

**2.2 SP2 data analysis**

**2.2.1 Aging degree of BC-containing particles**

Based on the rBC core mass ($m_{rBC}$) and scattering cross section ($C_s$) of the BC-containing particle derived from the SP2 measurements, the size of the BC-containing particle ($D_p$), including the rBC core and the coating materials, was calculated by Mie theory with a shell-and-core model (Zhang et al., 2016). In Mie calculation, the $D_p$ is retrieved from $C_s$, the size of rBC core ($D_c$) and the refractive indices of the non-BC shell ($RI_s$) and rBC core ($RI_c$).

The $RI_s$ value used in this study are 1.50-0i based on the chemical compositions of coating materials during the campaign period (Fig. S4 and the associated discussion in the supplementary information). In term of $RI_c$, we evaluated the sensitivity of $D_p$ values retrieved by Mie mode to the $RI_c$ values (Fig. S5 and the associated discussion in the supplementary information).

In the following calculation, the $RI_c$ of 2.26-1.26i was used (Taylor et al., 2015).
The $D_c$ is calculated using the $m_{rBC}$ and rBC core density ($\rho_c$, 1.8 g cm$^{-3}$ used in this study (Cappa et al., 2012)) assuming
a void-free sphere for rBC core, as given in Eq. (1). The size distribution of rBC cores under different pollution levels during
the campaign period is displayed in Fig. S6.
$$\boldsymbol{D_c} = (\frac{6m_{rBC}}{\pi\rho_c})^{1/3},$$ (1).
As in previous studies (Liu et al., 2013; Sedlacek et al., 2012; Zhang et al., 2016), the aging degree of BC-containing
particles was characterized by the $D_p/D_c$ ratio in this study. Higher $D_p/D_c$ ratio for BC-containing particles indicates higher
aging degree, i.e., more coating materials on the BC surface.
In terms of $D_p/D_c$ ratio of BC-containing particles, we focused on the rBC core size ($D_c$) above detection limit of SP2
incandescence ($D_c$ >75 nm), while the detection limit of SP2 scattering for the whole particle size ($D_p$) was not considered in
this study. If the BC-containing particles with rBC cores larger than ~75 nm is large enough to be detected by SP2 scattering
channel, we would calculated their whole particle size ($D_p$) using LEO method based on their scattering signal. If not, we
would assume that the $D_p$ was equal to the rBC core size ($D_c$). This assumption might lead to the underestimation of $D_p/D_c$
ratio of the BC-containing particles with size above the incandescence limit and blow the scattering limit. To evaluate the
uncertainty of $D_p/D_c$ ratio, we calculated the detect efficiency of SP2 scattering (Fig. S7 in the supplementary information). In
terms of BC-containing particles with rBC core larger than 75 nm (SP2 size cut for incandescence) observed in our site during
the campaign period, most of them (~90-100%, Fig. S7) exhibited particle size (180-500 nm shown in Fig. R8) larger than
SP2 size cut for scattering due to large coating materials on rBC cores. This indicated that the uncertainty of $D_p/D_c$ ratio
calculated in this study due to mismatch in the SP2 size cut for incandescence vs scattering is no more than 10%. High detection
efficiency of SP2 scattering for BC-containing particles observed in our site can be attribute to their large size (180-500 nm,
Fig. S8).
**2.2.2 BC optical properties**
Based on the size information on BC-containing particles (i.e., $D_c$ and $D_p$) obtained from the SP2 measurement (discussed in
Sec. 2.2.1), we used Mie theory with a shell-and-core model to retrieve the optical properties of BC-containing particles,
including the absorption enhancement ($E_{ab}$) of rBC, the mass absorption cross-section ($MAC$) of BC-containing particles and
bare rBC cores, the mass scattering cross section of bare BC core ($MSC_{core}$) and the absorption coefficient ($\sigma_{ab}$) of BC-
containing particles. The calculation of these parameters is described below.
The $E_{ab}$ characterizes the increase in BC light absorption due to the lensing effect of coating materials on the BC surface
and is used to quantify the theoretical light absorption capability of BC-containing particles in this study. The $E_{ab}$ is determined
by the ratio of the absorption cross section of the entire BC-containing particle ($C_{ab,p}$) to that of bare BC core ($C_{ab,c}$), as
expressed in Eq. (2):
$$E_{ab} = \frac{C_{ab,p}\ (D_c, D_p, RI_s, RI_c)}{C_{ab,c}(D_c, RI_c)},$$  (2)
where $C_{ab,p}$ is determined by the $D_c$, $D_p$, $RI_s$ and $RI_c$ using Mie calculation, and $C_{ab,c}$ is determined by the $D_c$ and $RI_c$.
The $MAC$ of BC-containing particles ($MAC_p$) and bare rBC cores ($MAC_c$) is defined as the $C_{ab,p}$ and $C_{ab,c}$ per unit rBC mass,
as Eqs. (3) and (4), respectively:
$$MAC_p = \frac{C_{ab,p}\ (D_c, D_p, RI_s, RI_c)}{m_{rBC}},$$  (3)
$$MAC_c = \frac{C_{ab,c}\ (D_c, RI_c)}{m_{rBC}},$$  (4)
The $MSC_{core}$ is the scattering cross sections of bare rBC cores ($C_{sca,c}$) per unit rBC mass, as calculated by Eq. (5):
$$MSC_{core} = \frac{C_{sca,c}\ (D_c, RI_c)}{m_{rBC}}.$$  (5)
The $\sigma_{ab}$ of BC-containing particles is calculated based on the $MAC$ and the rBC mass concentration ($C_{rBC}$) measured by the
SP2, as expressed in Eq. (6). The uncertainties of $\sigma_{ab}$ related to $MAC$ of bare rBC cores from Mie calculation was evaluated in
the supplementary information (Fig. S9 and the associated discussion).
$$\sigma_{ab,calculated} = MAC_p \times C_{rBC} = MAC_c \times E_{ab} \times C_{rBC},$$  (6).
**2.3 BC effective emission intensity**
To evaluate the impact of regional transport on BC-containing particles, we used a variant of the "effective emission intensity"
(EEI) defined by Lu et al. (2012) to quantify the amounts of BC over the observation site from different source regions. In this
study, the spatial origin of the BC observed at our site was divided into local sources in Beijing and regional sources in other
areas (i.e., Hebei, Tianjin, Shanxi and Inner Mongolia, Fig. S1). The EEI takes into account emission, transport, hydrophilic-
to-hydrophobic conversion, and removal processes (i.e., dry and wet deposition) of BC throughout the whole atmospheric
transport process from the origin of the BC emission to the receptor site. A novel back-trajectory approach was developed by
Lu et al. (2012) to calculate EEI values.
In this study, the back-trajectory analysis was performed by the Hybrid Single-Particle Lagrangian Integrated Trajectory
(HYSPLIT) model to obtain the transport pathways of BC to the observation site (40º00'17" N, 116º19'34" E) during the
campaign period (November 17-30, 2014). The 72-h back-trajectory at 100 m at every hour was calculated with the
meteorological fields of NCEP GDAS at a 1º×1º resolution. An anthropogenic BC emission inventory of China in the year
2012 at a resolution of 0.25º×0.25º was used to support the back-trajectory analysis. The gridded BC emission data are from
the Multi-resolution Emission Inventory for China (MEIC) developed by Tsinghua University (http://www.meicmodel.org).
We calculated the EEI of BC at a resolution of 0.25º×0.25º based on the algorithm developed by Lu et al. (2012). Following
a trajectory $l$ at every hour, the fresh BC emitted from a series of spatial grids in sequence (i.e., $l_1$, $l_2$, ..., $l_{i,...}$) are transported
to the receptor grid (i.e., $l_n$). The EEI of trajectory $l$ at the surface grid point $i$ ($EEI_{i,l}$) was determined by Eq. (7):
$$EEI_{i,l} = E_i \times TE_{i,l},$$  (7)
where $E_i$ is the BC emission at the surface grid point $i$, and $TE_{i,l}$ represents the BC transport efficiency of trajectory $l$ from the
grid point $i$ to the receptor site, as calculated by Eqs. (1)-(4) in Lu et al. (2012).

3        The total EEI of trajectory $l$ characterizes the total amount of BC transported to the observation site at every hour, expressed

as Eq. (8):
$EEI_{total} = \sum_{i=1}^{n} EEI_{i,l}$,                                                                        (8).
**2.4 BC radiative efficiency**
In this study, we use a parameter of the simple forcing efficiency (*SFE*) to roughly evaluate the radiative forcing of BC-
containing particles. The *SFE* is defined as normalized radiative forcing by BC mass, which is wavelength-dependent (Bond
and Bergstrom, 2006; Chen and Bond, 2010; Chylek and Wong, 1995; Saliba et al., 2016). The wavelength-dependent *SFE*
for BC-containing particles is determined by Eq. (9):
$\frac{dSFE}{d\lambda} = -\frac{1}{4}\frac{dS(\lambda)}{d\lambda}\tau_{atm}^2(\lambda)(1 - F_c) \times [2(1 - \alpha_s)^2\beta(\lambda) \cdot MSC_{core}(\lambda) - 4\alpha_s \times MAC(\lambda)]$,                  (9)
in which a wavelength ($\lambda$) of 550 nm is used in this study; dS ($\lambda$)/d$\lambda$ is the spectral solar irradiance, the value of which is from
the ASTM G173-03 Reference Spectra (1.86 W m$^{-2}$ nm$^{-1}$ at 550 nm); and the parameters $\tau_{atm}$, $F_c$, $\alpha_s$ and $\beta$ are the atmospheric
transmission (0.79), cloud fraction (0.6), urban surface albedo (0.15) and backscatter fraction (0.17), respectively (Jeong et al.,
2013; Chen and Bond, 2010; Park et al., 2011; Saliba et al., 2016).
**3 Results**
**3.1 Light absorption of BC-containing particles during the campaign period**
Figure 1 shows the time series of the PM$_1$ and rBC mass concentration, the diameter of BC-containing particles ($D_p$) and the
measured and calculated light absorption coefficient ($\sigma_{ab}$) at 880 nm. During the campaign period, four episodes with different
PM$_1$ evolution processes (Fig. 1a) were observed: November 18-21, November 23-24, November 25-26 and November 28-
30. The hourly PM$_1$ mass concentration ranged from 3.5-275 μg m$^{-3}$, with an average value of 91 μg m$^{-3}$ during the observed
period. The rBC mass concentration accounted for ~5% of PM$_1$. The $D_p$ including rBC cores and coating materials shown in
Figure 1(b) exhibited an excellent temporal coherence with the rBC mass concentration. Figure 1(b) shows that the number
distribution of $D_p$ for BC-containing particles exhibited a peak at 180-320 nm, significantly larger than the peak value ($D_c$ of
~95 nm) for number size distribution of bare rBC cores (Fig. S5a) due to larges of coating materials on BC surface. The size
information (namely entire particle size and rBC core size) of BC-containing particles observed in our study was consistent
with those ($D_p$ of ~200-300 nm and $D_c$ of ~70-100 nm) in previous studies in China (Gong et al., 2016; Huang et al., 2012;
Wang et al., 2014). Moreover, the $D_p$ exhibited sustained growth from ~180 nm to ~400 nm during a pollution episode, which
could be a consequence of the increase in either $D_c$ or coating materials, or both. Figure S6a shows a slight change in $D_c$ with

pollution development. However, the coating thickness of BC-containing particles increased with $PM_1$ concentration (Fig. S10a). Therefore, the sustained growth of $D_p$ during a pollution episode was dominated by more coating materials under more polluted conditions. Figure S10 shows the simultaneous increase in the rBC mass concentration and the amount of coating materials on the BC surface, which could significantly enhance the light absorption of BC-containing particles. Figure S11 shows that the measured absorption coefficient ($\sigma_{ab, measured}$) at 880 nm (dominated by BC component at this wavelength (Drinovec et al., 2015)) for the aerosol particles measured by the AE33 exhibited an 18-fold increase in conjunction with an increase in the $PM_1$ concentration from ~10 μg m$^{-3}$ to ~230 μg m$^{-3}$, larger than the increase (~14 times) of rBC mass concentration.

To valid SP2 measurements and Mie calculation used in this study, we compared the calculated light absorption coefficient ($\sigma_{ab, calculated}$) of BC-containing particles using Eq. (6) with the measured light absorption coefficient ($\sigma_{ab, measured}$) from AE33, as shown in Fig. 1c. The $\sigma_{ab, calculated}$ values for BC-containing particles showed an excellent agreement with the $\sigma_{ab, measured}$ values measured by the AE33, with a difference of ~10% ($R^2$=0.98). The difference was dominated by the uncertainties from compensation algorithm used in the AE33 measurements (~15%, details shown in Fig. S2 and the associated discussion in the supplementary information) and Mie calculation (10-20%, Fig. S4, Fig. S5 and Fig. S9 and the associated discussion in the supplementary information). The uncertainty evaluation revealed that the difference between $\sigma_{ab,calculated}$ and $\sigma_{ab,cmeasured}$ (~10%) shown in Fig. 1c is reasonable. The comparison between the $\sigma_{ab, calculated}$ and $\sigma_{ab, measured}$ values implied that the optical properties (i.e., $\sigma_{ab,}$, $MAC$ and $E_{ab}$) of the BC-containing particles derived from Mie calculation combining with SP2 measurements (i.e., rBC concentrations, $D_p$ and $D_c$) were reliable in our case.

**3.2 Enhancement of the theoretical light absorption capability of black carbon associated with air pollution**

**3.2.1 The $D_p/D_c$ ratio and calculated $E_{ab}$ under different $PM_1$ concentrations**

Previous theoretical studies reported that the coating materials on the BC surface can significantly enhance the light absorption of BC via the lensing effect (Fuller et al., 1999; Jacobson, 2001; Lack and Cappa, 2010; Moffet et al., 2009). In other words, the aging degree of BC-containing particles (characterized by the $D_p/D_c$ ratio in this study) determines their theoretical light absorption capability (characterized by the calculated $MAC$ and $E_{ab}$ in this study). However, whether and how the aging degree and light absorption capability of BC-containing particles will change under different pollution levels remains unclear. During the campaign period, we found that the mass-averaged values of the $D_p/D_c$ ratio and calculated $E_{ab}$ of BC-containing particles increased with increasing air pollution levels.

Figure 2a shows the $D_p/D_c$ ratio and calculated $E_{ab}$ of BC-containing particles with rBC cores at 75-300 nm under different $PM_1$ concentrations in the range of 1.2-3.5 and 1.3-3.1, respectively. In terms of BC-containing particles with a certain rBC core size, their $D_p/D_c$ ratio and calculated $E_{ab}$ were greater under higher $PM_1$ concentrations, which could be attributed to more coating materials on BC surface under more pollution environment. The increase of both primary and secondary components under more polluted conditions was favorable to BC aging by coagulation and condensation, which happen mostly between

BC and non-BC species. On average (i.e., mass-weighted mean across rBC core size larger than ~75 nm), the $D_p/D_c$ and calculated $E_{ab}$ for observed BC-containing particles in SP2 under different $PM_1$ concentrations during the campaign period varied in the range of 1.6-2.2 and 1.6-2.0, respectively (Fig. 2b). Correspondingly, the mass-averaged values of the $D_p/D_c$ and calculated $E_{ab}$ of BC-containing particles increased by ~33% and ~18%, respectively, with increasing $PM_1$ concentrations from ~10 μg m$^{-3}$ to ~230 μg m$^{-3}$ (companied with rBC mass concentration increasing from ~0.7 μg m$^{-3}$ to ~11 μg m$^{-3}$).

Figure 3 shows the increase of the $D_p/D_c$ ratio and calculated $E_{ab}$ ($IR_{Dp/Dc}$ and $IR_{Eab}$, respectively) for BC-containing particles with increasing $PM_1$ concentrations form 10 μg m$^{-3}$ to 230 μg m$^{-3}$ as a function of rBC core size. Based on the $D_p/D_c$ ratio and calculated $E_{ab}$ of BC-containing particles with size-resolved rBC cores in SP2 measurement under different $PM_1$ concentration (shown in Fig. 2a), we can obtain the measured $IR_{Dp/Dc}$ and $IR_{Eab}$ as a function of rBC core size. When $PM_1$ concentration increasing from ~10 μg m$^{-3}$ to ~230 μg m$^{-3}$ during the campaign period, the $D_p/D_c$ ratio and calculated $E_{ab}$ of BC-containing particle with rBC cores at 75-200 nm increased by 28-48% and 13-36%, respectively. The size-dependent increase of $D_p/D_c$ ratio and calculated $E_{ab}$ associated with air pollution indicated that the aging process of smaller rBC was relatively more sensitive to air pollution levels. This could be attributed to the fact that the condensational growth associated with air pollution due to the formation of secondary components is more effective for smaller particles in terms of increasing the diameter (Metcalf et al., 2013).

Meanwhile, following a semiquantitative analysis using in Metcalf et al. (2013), we calculated the $IR_{Dp/Dc}$ and $IR_{Eab}$ based on diffusion-controlled growth law (Seinfeld and Pandis 2006). The calculated $IR_{Dp/Dc}$ and $IR_{Eab}$ would compare with the measured ones from SP2.

According to the diffusion-controlled growth law (Seinfeld and Pandis 2006), the evolution of the size of BC-containing particles ($D_p$) is shown:

$$\frac{dD_P}{dt} = \frac{A}{D_P} \qquad (10)$$

in which, $\frac{dD_P}{dt}$ represents the diffusion-controlled growth rate; $A$ is a parameter. Integrating Eq. (10) with $D_p$ ($t = 0$) = $D_c$:

$$D_p^2 = D_c^2 + 2At = D_c^2 + B \qquad (11)$$

in which, $D_c$ is rBC core diameter; $B$ (i.e., equal to $2At$) is a parameter, varying under different $PM_1$ concentrations.

Following Eq. (11), the $D_p/D_c$ ratio is given:

$$\frac{D_p}{D_c} = (\frac{B}{D_c^2} + 1)^{1/2} \qquad (12)$$

where the parameter $B$ is determined by the value of the measured $D_p/D_c$ ratio with $D_c$ of 160 nm under different $PM_1$ concentrations.

The $IR_{Dp/Dc}$ for BC-containing particles with $PM_1$ concentration increasing from 10 μg m$^{-3}$ to 230 μg m$^{-3}$ can be calculated by Eq. (13):

$$IR_{Dp/Dc} = \frac{(\frac{D_p}{D_c})_{230} - (\frac{D_p}{D_c})_{10}}{(\frac{D_p}{D_c})_{10}} = \frac{(\frac{B_{230}}{D_c^2} + 1)^{1/2} - (\frac{B_{10}}{D_c^2} + 1)^{1/2}}{(\frac{B_{10}}{D_c^2} + 1)^{1/2}}$$
(13)

where $(\frac{D_p}{D_c})_{230}$ and $(\frac{D_p}{D_c})_{10}$ represent the $D_p/D_c$ ratio when $PM_1$ concentrations are 230 μg m$^{-3}$ and 10 μg m$^{-3}$, respectively; $B_{230}$
and $B_{10}$ are parameter $B$ with $PM_1$ concentrations of 230 μg m$^{-3}$ and 10 μg m$^{-3}$, respectively.

4        The $IR_{Eab}$ for BC-containing particles with $PM_1$ concentration increasing from 10 μg m$^{-3}$ to 230 μg m$^{-3}$ can be derived

based on $E_{ab}=k\times(D_p/D_c)$, as expressed in Eq. (5):

$$IR_{Eab} = \frac{k_{230} \times (\frac{D_p}{D_c})_{230} - k_{10} \times (\frac{D_p}{D_c})_{10}}{k_{10} \times (\frac{D_p}{D_c})_{10}} = \frac{k_{230} \times (\frac{B_{230}}{D_c^2} + 1)^{1/2} - k_{10} \times (\frac{B_{10}}{D_c^2} + 1)^{1/2}}{k_{10} \times (\frac{B_{10}}{D_c^2} + 1)^{1/2}}$$
(14)

7        We compared the calculated $IR_{Dp/Dc}$ and $IR_{Eab}$ based on Eqs. (13) and (14) with those from SP2 measurements, as shown

in Fig.3. The agreement indicated that the increase of the $D_p/D_c$ and $E_{ab}$ for BC-containing particles with increasing $PM_1$
concentrations follow the diffusion-controlled growth law.
**3.2.2 Changing rate of the theoretical light absorption capability of black carbon**
Figure 4a explores the relationship between the changing rate of calculated $E_{ab}$ ($k_{Eab}$) and the changing rates of $PM_1$
concentrations ($k_{PM1}$) with pollution development. Linear relationships were estimated, i.e., $k_{Eab} \approx 0.051\ k_{PM1}$, revealing that
rapid increases in air pollution levels could lead to rapid increases in BC light absorption capability. When the $PM_1$
concentration exhibited a sharp increase related to an extreme haze episode (Zheng et al., 2015), the increase in BC light
absorption capability was dramatic.

16       Figure 4b shows frequency distribution of $k_{Eab}$, $k_{PM1}$, and $k_{Eab}/k_{PM1}$ ratio. During the campaign period, most of $k_{Eab}$ and

$k_{PM1}$ values were in the range of -50%-50% h$^{-1}$ and -4%-4% h$^{-1}$, respectively, revealing a lower changing rate for BC aging than
that for $PM_1$ concentration. The peak value of frequency distribution of $k_{Eab}$ was around zero, indicating the BC-containing
particles were shrinking as often as they were growing. The $k_{Eab}/k_{PM1}$ ratio characterized the sensitivity of the change of
calculated $E_{ab}$ with changing $PM_1$ concentrations. The frequency distribution of $k_{Eab}/k_{PM1}$ ratio showed that ~60% values were
in the range of 0-1, with a peak value around 0.05. Smaller values of $k_{Eab}/k_{PM1}$ ratio indicated that the change of calculated $E_{ab}$
was not sensitive to variations in $PM_1$ concentrations .The growth rate of $E_{ab}$ (for BC samples with $k_{Eab}>0$: 0.1-7.3% h$^{-1}$)
observed at our study was consistent with the BC aging rate (0.2-7.8% h$^{-1}$) in previous studies (Cheng et al., 2012; Moteki et
al., 2007; Shiraiwa et al., 2007).

25       Moreover, we found that the growth rate of $E_{ab}$ decreased with increasing $PM_1$ mass concentrations (Fig. S12a), indicating

that the increase in the theoretical light absorption capability of BC-containing particles slowed with further pollution
development. This can be explained by larger BC-containing particles when $PM_1$ concentration is higher (Fig. S12b). The net
change in diameter for a given amount of material deposited decreases with increasing particle size due to surface-to-volume
scaling, which would expect the growth rate of particles to decrease with increasing $PM_1$ concentration and thus the $k_{Eab}$ would
also decrease (Fig. S12a).
The evolution of theoretical light absorption of BC with pollution levels depends on the change in both rBC mass
concentrations and calculated $E_{ab}$. Figure S13 shows markedly smaller $k_{Eab}$ than the changing rate of rBC mass concentrations
($k_{rBC}$) (i.e., $k_{Eab} \approx 0.027\ k_{rBC}$), indicating the change of calculated $E_{ab}$ was significantly slower than that of rBC mass
concentrations under different pollution levels. Due to less sensitive for calculated $E_{ab}$ to change in air pollution levels
compared with that for rBC mass concentrations, some previous measurements (McMeeking et al., 2011; Ram et al., 2009;
Wang et al., 2014b; Andreae, et al., 2008) would not have been able to discern a difference of $E_{ab}$ easily among different
pollution levels and thus just focus on the change of BC mass concentration. This would lead to uncertainties in estimation of
BC light absorption.  In our case, Figure. 2b reveals the mass-weighted average of calculated $E_{ab}$ increased by ~18% with $PM_1$
concentration increasing from ~10 μg m$^{-3}$ to ~230 μg m$^{-3}$ (companied with rBC mass concentration increasing from ~0.7 μg
m$^{-3}$ to ~11 μg m$^{-3}$).  If the increase in calculated $E_{ab}$ of BC with $PM_1$ increase was neglected in this study, the theoretical light
absorption of BC-containing particles would be underestimated by ~18% under polluted conditions.
**3.3 Contribution of regional transport**
BC aging in the atmosphere, namely BC internally mixing with other aerosol components, is associated with atmospheric
transport (Gustafsson and Ramanathan, 2016). In Beijing, the rapid increase in aerosol particle concentrations during pollution
episodes is most likely caused by regional transport of polluted air mass (Yang et al., 2015; Zheng et al., 2015). Therefore,
regional transport of pollution may play an important role in the enhancement of BC light absorption capability associated
with air pollution. In this study, we used the EEI analysis (Lu et al., 2012) to explore the effects of regional transport on the
increase in theoretical light absorption capability of BC with increasing pollution levels.
Figure 5 shows the spatial distribution (0.25°×0.25°) of the EEI for BC transported to the observation site under different
pollution levels (i.e., clear, slightly polluted and polluted periods) during the campaign period. The spatial origin of the BC
observed at our site varied significantly among the different pollution periods. In this study, the spatial origin of total BC in
the site was classified into local Beijing and other regions (i.e., outside of Beijing, considered as regional origins in this study).
Noted that the local region (i.e., Beijing) defined in this study is smaller than areas outside of Beijing (e.g., Hebei, Tianjin,
Shanxi and Inner Mongolia (Fig. S1)). Table 1 lists the contribution of BC from regional origins (i.e., $EEI_{ousiede}/EEI_{total}$ ratio).
During polluted period, the contributions of BC from regional origins was ~63%, larger than that from local Beijing (~37%).
This was partly due to comparing the contributions from a small region (Beijing) and a large region (outside of Beijing). In
this study, we focus on comparing the contributions of BC from outside of Beijing (considered as regional origins in this study)
among different pollution levels (i.e., clean, slight polluted and polluted period). The BC from regional origins (i.e., outside of
Beijing)  accounted for ~21%, 39% and ~63% of total BC amount in the site during the clean, slightly polluted and polluted
periods, respectively. This revealed that the regional contribution to BC over Beijing increased as the air pollution levels
increased.

2       Due to the increase of the regional contribution, the total BC amount transported to the observation site, characterized by

the EEI *(EEI*$_{total}$) in this study, increased under more polluted condition. Table 1 shows that the *EEI*$_{total}$ was 4.6 times higher
during the polluted period than during the clean period, revealing that polluted air mass brought more BC to Beijing. BC
concentration in the site strongly depends on both total BC amount (transported from local Beijing and other regions,
characterized by *EE*I$_{total}$ in this study) and local meteorology. Table 1 shows that the BC concentrations from the clean period
to the polluted period increase by ~7.4 times. The increase of *EEI*$_{total}$ (~4.6 times) accounted for ~62% the increase in BC mass
concentrations (~7.4 times). This indicated that the adverse local meteorology contributed ~38% of the increase in BC mass
concentration in the site from the clean period to the polluted period. Compared with regional transport, less effect of adverse
local meteorology might be attributed to relatively small areas defined as the local region (i.e., Beijing) in this study. Polluted
events in China always occur over a large region, e.g., North China Plain (Yang et al., 2017; Zheng et al., 2015). For our case,
the adverse meteorology during polluted days in the whole large region including Beijing and other areas can lead to the
increase of pollutants and then more transport of pollutants into Beijing. Yang et al. (2017) found that the increases in BC
concentration under polluted conditions over the North China Plain (including Beijing and other adjacent areas) is dominated
by its local emissions due to adverse meteorology.
Under different pollution levels, regional transport not only influenced the BC mass concentrations but also the BC aging
process and timescale. Table 1 shows that the mass-average value of the $D_p/D_c$ ratio of BC-containing particles at our site was
~2.04 during the polluted period, significantly higher than that observed during the clean period (~1.62) and slightly polluted
period (~1.81). On one hand, under more polluted conditions, more BC-containing particles in Beijing were from regional
sources and thus had undergone a longer aging time during the transport than the BC from local sources in Beijing. On the
other hand, compared with the BC carried in the clean air mass from the northwest of Beijing during the clean period (Fig.
5a), the BC-containing particles in the polluted air mass undergoing regional transport from the region south of Beijing (i.e.,
Hebei, one of the most polluted provinces in China with high pollutant emission) during the polluted periods (Fig. 5c). Peng
et al. (2015) pointed out higher BC aging rates under more polluted environments, indicating that BC-containing particles
passing though polluted regions would show higher aging rates during atmospheric transport than that from clean regions. The
mass-average values of calculated $E_{ab}$ for BC-containing particles observed at our site were ~1.66, ~1.81 and ~1.91 during the
clean, slightly polluted and polluted periods, respectively (Table 1), showing that the theoretical light absorption capability of
BC-containing particles observed at our site increased with increasing regional contributions. Our results demonstrated the
importance of regional transport in the enhancement of BC light absorption capability associated with air pollution.
To further explore the importance of aging during regional transport, its contributions were compared with those of local
chemical processes with respect to increases in $D_p/D_c$ ratio and theoretical light absorption capability of BC-containing
particles associated with air pollution. Considering the importance of photochemical oxidation in coating formation on BC
surface (Metcalf et al., 2013; Peng et al., 2015), we evaluated the contribution of local photochemical production by the
changes of O$_3$ concentrations in the atmosphere. On the other hand, the changes in the amount of BC from regional transport

was characterized by variation of $EEI_{total}$, which was used to evaluate the contributions of regional transport to BC aging..

Figure 5a shows that the $EEI_{total}$ per hour exhibited a temporal coherence with the mass-averaged values of the $D_p/D_c$ ratio and calculated $E_{ab}$ of BC-containing particles. In contrast, the $O_3$ concentrations showed a different temporal trend. When $PM_1$ concentrations were higher than ~120 μg m$^{-3}$, $O_3$ concentrations decreased to ~2 ppb. Zheng et al. (2015) has demonstrated the weakened importance of photochemistry in the production and aging of secondary aerosols in Beijing under polluted conditions due to decrease of oxidant concentrations. This indicated that the photochemical processing in BC aging may be weakened under higher polluted levels (i.e., $PM_1$>120μg m$^{-3}$). Noted that photochemical processing is not the only possible pathway in BC aging process and other pathways were not discussed in this study. The local aging process of BC might be enhanced by other pathways. For example, high concentrations of aerosols under polluted environment may compensate the adverse photochemical conditions for BC aging. In summary, the increases in the aging degree and theoretical light absorption capability of BC-containing particles with increasing air pollution were more likely caused by aging during regional transport than by local photochemical production.

According to the evolution of the $EEI_{total}$ values and $O_3$ concentrations with increasing air pollution levels (Fig. 5b1 and Fig.5b2), we separated the pollution levels into two periods. When $PM_1$ concentrations were lower than ~120 μg m$^{-3}$ and rBC mass concentrations were lower than ~6 μg m$^{-3}$, the normalized $EEI_{total}$ increased from ~3 to ~18 with increasing air pollution levels, and the $O_3$ concentrations decreased from ~20 ppb to ~2 ppb, indicating enhanced regional contributions and weakened local photochemical production at observation site. In this period, Fig. 5b3 and Fig. 5b4 show that the mass-averaged values of calculated $E_{ab}$ and the $D_p/D_c$ ratio of BC-containing particles increased from ~1.6 to ~1.9 and from ~1.6 to ~2.0, respectively, with the increase in the normalized $EEI_{total}$ (from ~3 to 18) and the decrease in the $O_3$ concentrations (from ~20 to 2 ppb). Therefore, in terms of the increase in the BC light absorption capability with increasing air pollution levels, this period (i.e., conditions of $PM_1$<120 μg m$^{-3}$ and rBC < 6 μg m$^{-3}$) represented a regional transport-controlled period. The increase in calculated $E_{ab}$ of BC-containing particles (~1.6-1.9) during this regional transport-controlled period accounted for ~75% of the increase in calculated $E_{ab}$ (~1.6-2.0) with increasing air pollution during the whole campaign period. Therefore, the aging process during the regional transport dominated the increase in the theoretical light absorption capability of BC-containing particles in Beijing during the campaign period. Another period is defined by $PM_1$ concentrations of more than ~120 μg m$^{-3}$ and rBC mass concentrations of more than ~6 μg m$^{-3}$, during which both the $EEI_{total}$ and $O_3$ concentrations showed slight changes with increasing air pollution levels. In this period, the increase in calculated $E_{ab}$ of BC-containing particles (from ~1.9 to ~2.0) might be attributed to local chemical production (e.g., heterogeneous reaction) in Beijing.

**3.4 Implications for BC radiative forcing**

The increase in BC light absorption capability with increasing air pollution levels suggests that greater solar absorption (i.e., direct radiative forcing (DRF)) by atmospheric BC-containing particles occurs under more polluted conditions. The DRF of atmospheric BC-containing particles depends not only on the BC mass concentrations but also on the BC forcing efficiency,

which strongly depends on the light absorption capability of BC-containing particles. In this study, the forcing efficiency of BC-containing particles was estimated based on a simple radiation transfer model (Eq. (9)). Figure 7 shows that with increasing pollution levels (i.e., $PM_1$ increasing from ~10 μg m$^{-3}$ to ~230 μg m$^{-3}$) during the campaign period, the mass-averaged values of calculated MAC at 550 nm for BC-containing particles increased from ~11 m$^2$ g$^{-1}$ to ~14 m$^2$ g$^{-1}$, which resulted in the SFE of BC-containing particles increasing from ~0.7 m$^2$ g$^{-1}$ nm$^{-1}$ to ~0.9 m$^2$ g$^{-1}$ nm$^{-1}$. This revealed that the enhanced light-absorption capability of BC-containing particles under more pollution environment would increase their radiative forcing.

Considering the increase in both of the light-absorption capability (~18%) and mass concentrations (from ~0.7 μg m$^{-3}$ to ~11 μg m$^{-3}$) of BC when $PM_1$ increased from ~10 μg m$^{-3}$ to ~230 μg m$^{-3}$, the DRF of BC-containing particles increased from ~0.3 W m$^{-2}$ to ~6.4 W m$^{-2}$ (i.e., DRF$_{with}$ shown in Fig. 7c and 7d). This revealed the importance of BC in terms of solar absorption under more polluted conditions. However, when the theoretical DRF of BC-containing particles was derived without considering the light-absorption enhancement caused by coating materials (i.e., DRF$_{without}$ shown in Fig. 7c and 7d), it varied in the range of 0.2-3.3 W m$^{-2}$, which only depended on the mass concentrations of BC under different pollution levels. The DRF$_{witht}$ was significantly higher than DRF$_{without}$, identifying the amplification of DRF caused by coating materials on the BC surface. More significant difference between DRF$_{with}$ and DRF$_{without}$ was found under more polluted environment, which was due to the increase of BC light-absorption capability with increasing pollution levels. This indicated that more important for the coating materials to the direct radiative forcing of BC under more polluted environment.

The enhanced climate effects of BC aerosols in Beijing could be taken to be representative of polluted regions in China. Previous measurements of BC aerosols in China (Zheng et al., 2015; Wang et al., 2014b; Zhao et al., 2017; Gong et al., 2016; Huang et al., 2013; Andreae et al., 2008; Zhang et al., 2014) showed that the BC mass concentrations in different regions (e.g., Beijing, Xi'an, Nanjing, Shanghai and Guangzhou) reached values of ~10-50 μg m$^{-3}$ during polluted periods (Table S2), similar to our measurements. Therefore, our BC aerosol observations in Beijing were not a special case. In China, high concentrations of BC aerosols under polluted conditions always occur on a regional scale due to intense BC emissions (Zhang et al., 2009; Li et al., 2017) and significant regional transport (Zheng et al., 2015; Wang et al., 2014a; Zhao et al., 2013). Our findings in Beijing can provide some implication in the difference of BC radiative forcing in other regions among different air pollution levels.

**4 Discussion: breaking the amplification effect by emission control**

Our results reveal that under more polluted environment, the BC-containing particles are characterized by more BC mass concentrations and more coating materials on BC surface and therefore higher light absorption capability. As shown in Fig. 8, this amplification effect on BC light absorption associated with air pollution is caused by increasing BC concentration and at the same time enhanced light absorption capability of BC-containing particles by more coating production in the more polluted air. Variation of both the mass concentration and light absorption capability of BC associated with air pollution strongly depend on the air pollutant emission (e.g., BC, $SO_2$, $NO_x$ and VOC). Under polluted environment, polluted air mass from high emission

areas not only brings more BC, but also more coating materials on BC surface due to more precursors of secondary
components.
Air pollution control measures may, on the other hand, break this amplification effect by reducing BC concentration and
at the same time lowering the light absorption capability of BC-containing particles by slowing down the coating processes
with a cleaner air (Fig. 7). Take air pollution controls during the 2014 Asia-Pacific Economic Cooperation meeting (APEC) in
Beijing, China as an example, we found that as a result of emission controls on local Beijing and areas adjacent to Beijing
(i.e., Hebei, Tianjin, Shanxi, Henan, Shandong and Inner Mongolia), light absorption of BC-containing particles decreased
significantly during APEC compared to that of before APEC (Zhang et al., 2018). This is not only contributed by a reduction
of BC mass concentration, but also by lower light absorption capability of BC-containing particles with less coating materials
on BC surface in cleaner atmosphere conditions, indicating that synergetic emission reduction of multi-pollutants could
achieve co-benefits of both air quality and climate.
**5 Conclusions**
The light absorption of BC-containing particles depends not only on the BC mass concentration but also on their light
absorption capability (characterized by the calculated $MAC$ and $E_{ab}$ in this study). In this work, we investigated the difference
of theoretical light absorption capability of BC under different air pollution conditions. During an intensive field measurement
campaign in Beijing, China, we found that with increasing pollution levels, the increase in BC mass concentration was always
accompanied by an increase in the theoretical light absorption capability of BC, resulting an amplification effect on the light
absorption of the ambient BC-containing particles. During the campaign period, the hourly values of mass-weighted averages
of the $D_p/D_c$ ratio and calculated $E_{ab}$ for BC-containing particles was in the range of 1.5-2.3 and 1.5-2.0, respectively. When
PM$_1$ concentration increased from ~10 μg m$^{-3}$ to ~230 μg m$^{-3}$ accompanied with the rBC mass concentration in the range of
~0.7-11 μg m$^{-3}$, the mass-weighted averages of the $D_p/D_c$ ratio and calculated $E_{ab}$ values increased by ~33% and ~18%,
respectively. The increase in BC light absorption capability associated with increasing air pollution can be explained by the
increase in coating materials on the BC surface under more polluted conditions. Moreover, the increase of the $D_p/D_c$ ratio and
calculated $E_{ab}$ with increasing air pollution levels was size-dependent, namely more increase was exhibited for smaller rBC
cores. This indicated that the aging degree and light absorption capability of smaller rBC was more sensitive to air pollution
levels. Using a semiquantitative analysis method based on the diffusion-controlled growth law (Seinfeld and Pandis 2006), we
also calculated the theoretical increase in the $D_p/D_c$ ratio and calculated $E_{ab}$ with increasing pollution levels for BC-containing
particles with size-resolved rBC cores. The agreement between the measured and theoretical increase in the $D_p/D_c$ ratio and
$E_{ab}$ indicated the increase of coating materials on BC surface with increasing PM$_1$ concentrations following the diffusion-
controlled growth law.
The relationships between the changing rate of calculated $E_{ab}$ ($k_{Eab}$) with air pollution development and that of the PM$_1$ and
rBC concentrations ($k_{PM1}$ and $k_{rBC}$, respectively) were estimated: $k_{Eab} \approx 0.051\ k_{PM1}$, and $k_{Eab} \approx 0.027\ k_{rBC}$. During the campaign

period, $k_{Eab}$ values were in the range of -4%-4% h$^{-1}$, with a peak of frequency distribution around zero, indicating that the BC-containing particles were shrinking as often as they were growing. The frequency distribution of $k_{Eab}/k_{PM1}$ ratio showed that a peak value around 0.05, revealing that the change of calculated $E_{ab}$ was not sensitive to variations in PM$_1$ concentrations. Although the changing rate of the BC light absorption capability was significantly lower than that of the BC mass concentration, the effect of enhanced BC light absorption capability on the light absorption of ambient BC-containing particles under polluted conditions is not negligible. In our case, if we had not considered the increase in the theoretical light absorption capability of BC with increasing air pollution during the campaign period, the theoretical light absorption of BC-containing particles under polluted conditions would have been underestimated by ~18%.

The increase in BC light absorption capability with increasing pollution levels in Beijing was controlled by aging during regional transport. The EEI analysis showed that ~63% of the BC observed at our site was transported from regional sources (i.e., areas outside of Beijing) during the polluted period, whereas the regional contributions were significantly lower (~21%) during the clean period. More BC in more polluted air from regional transport could lead to a higher local BC concentration in Beijing. Not only more BC but also more coatings are carried into Beijing by more polluted regional air mass (Fig. 8 (a)), which can be explained by more coating precursors (e.g. SO$_2$, NO$_x$ and VOC) in a more polluted air. Moreover, we separated the change of the $D_p/D_c$ ratio and theoretical light absorption capability of BC associated with air pollution into regional transport-controlled period (i.e., PM$_1$ < 120 μg m$^{-3}$ and BC < 6 μg m$^{-3}$) and local chemistry-controlled period (i.e., PM$_1$ > 120 μg m$^{-3}$ and BC > 6 μg m$^{-3}$). In the regional transport-controlled period, the mass-averaged values of $D_p/D_c$ and calculated $E_{ab}$ of BC-containing particles in Beijing increased from ~1.6 to ~2.2 and from ~1.6 to ~2.0, respectively, with increasing pollution levels. The further increase in mass-averaged values of $D_p/D_c$ (~ 2.0 to ~2.2) and calculated $E_{ab}$ (~1.9 to ~2.0) associated with air pollution is harder and is mostly likely attributed to local chemical production. Therefore, we attributed ~75% of the increase in theoretical light absorption capability of BC with increasing air pollution during the campaign period to aging during regional transport, demonstrating the regional transport has an important influence on the variations of light absorption capability of BC-containing particles in Beijing under different pollution levels.

Due to the increase in BC light absorption capability with increasing air pollution levels, stronger forcing efficiency of the BC-containing particles was found under more polluted conditions. During the campaign period, the BC forcing efficiency increased by ~18% with PM$_1$ increasing from ~10 μg m$^{-3}$ to ~230 μg m$^{-3}$. Considering the increasing in both of BC forcing efficiency and BC mass concentration, the DRF values of ambient BC-containing particles could increase from ~0.3 W m$^{-2}$ during the clean periods to ~6.4 W m$^{-2}$ during the polluted periods. The results identified that BC in more polluted environment exhibited a larger DRF radiative forcing, which was caused not only by the increase of BC mass concentrations but also by the enhancement of BC light absorption capability.

The amplification effect on BC DRF due to the increase of BC light absorption capability introduced in this work not only concerns in Beijing but is also likely to operate in other polluted regions in China The amplification effect not only could increase the direct contribution of BC to air pollution and climate change due to more light absorption, but also would enhance the indirect contribution by stronger aerosol-meteorology and aerosol-climate feedbacks. Our finds in this work can provide

some implication in the difference of BC-related effect on air quality and climate under different air pollution conditions (e.g., air clean and putted environment) due to change in BC light absorption capability associated with air pollution.

The air pollution control may break the amplification effect by reducing BC concentration and at the same time lower the light absorption capability of BC-containing particles by slowdown the coating processes with a cleaner air. Thereby, breaking the amplification effect by emission control would achieve a co-benefit effect by simultaneous mitigation of air pollution and climate change. Further study will focus on if and how emission reduction of BC and other pollutants in China will break the amplification effect.

## Acknowledgments

This work was supported by the National Natural Science Foundation of China (41625020, 41571130032 and 91644218) and the Guangdong "Pearl River Talents Plan" (2016ZT06N263).

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

**Table 1.** The average PM$_1$ mass concentration, rBC mass concentration, normalized $EEI_{total}$, $EEI_{outside}/EEI_{total}$ ratio, the mass-
averaged values of the $D_p/D_c$ ratio and calculated $E_{ab}$ during clean, slightly polluted and polluted periods. $EEI_{outside}$ is the EEI
of BC from areas outside Beijing; the $EEI_{ouside}/EEI_{total}$ ratio reflects the amount of BC contributed by regional transport to the
total amount of BC observed at our site. The clean (PM$_{2.5}$ ≤ 35 μg m$^{-3}$), slightly polluted (35 μg m$^{-3}$ < PM$_{2.5}$ ≤ 115 μg m$^{-3}$) and
polluted (PM$_{2.5}$ > 115 μg m$^{-3}$) periods were classified according to the Air Quality Index (Zheng et al. 2015).

|  | Clean | Slightly polluted | Polluted |
|---|---|---|---|
| PM$_1$ (μg m$^{-3}$) | 12.57 | 54.26 | 141.93 |
| rBC (μg m$^{-3}$) | 0.82 | 2.89 | 6.07 |
| Normalized $EEI_{total}$ | 3.68 | 9.19 | 16.87 |
| $EEI_{outside}/EEI_{total}$ | 0.21 | 0.39 | 0.63 |
| $D_p/D_c$ | 1.62 | 1.81 | 2.04 |
| Calculated $E_{ab}$ | 1.66 | 1.81 | 1.91 |

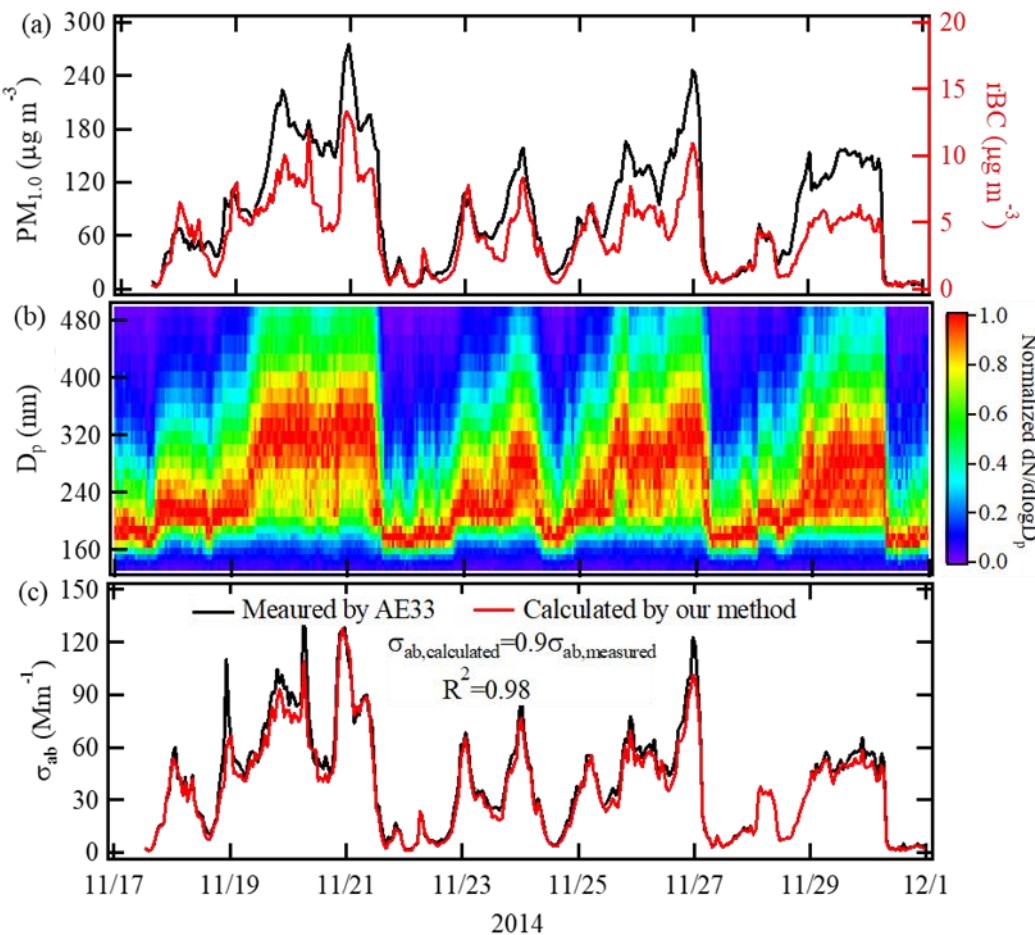

2    **Figure 1.** Time series of (a) the PM$_1$ and rBC mass concentrations, (b) the diameter of BC-containing particles ($D_p$) and (c)

3    the light absorption coefficient ($\sigma_{ab}$) at 880 nm. The correlation between the calculated $\sigma_{ab}$ ($\sigma_{ab,\,calculated}$) using Mie theory

4    combined with SP2 measurements and the measured $\sigma_{ab}$ ($\sigma_{ab,\,measured}$) by the AE33 is also shown in (c).

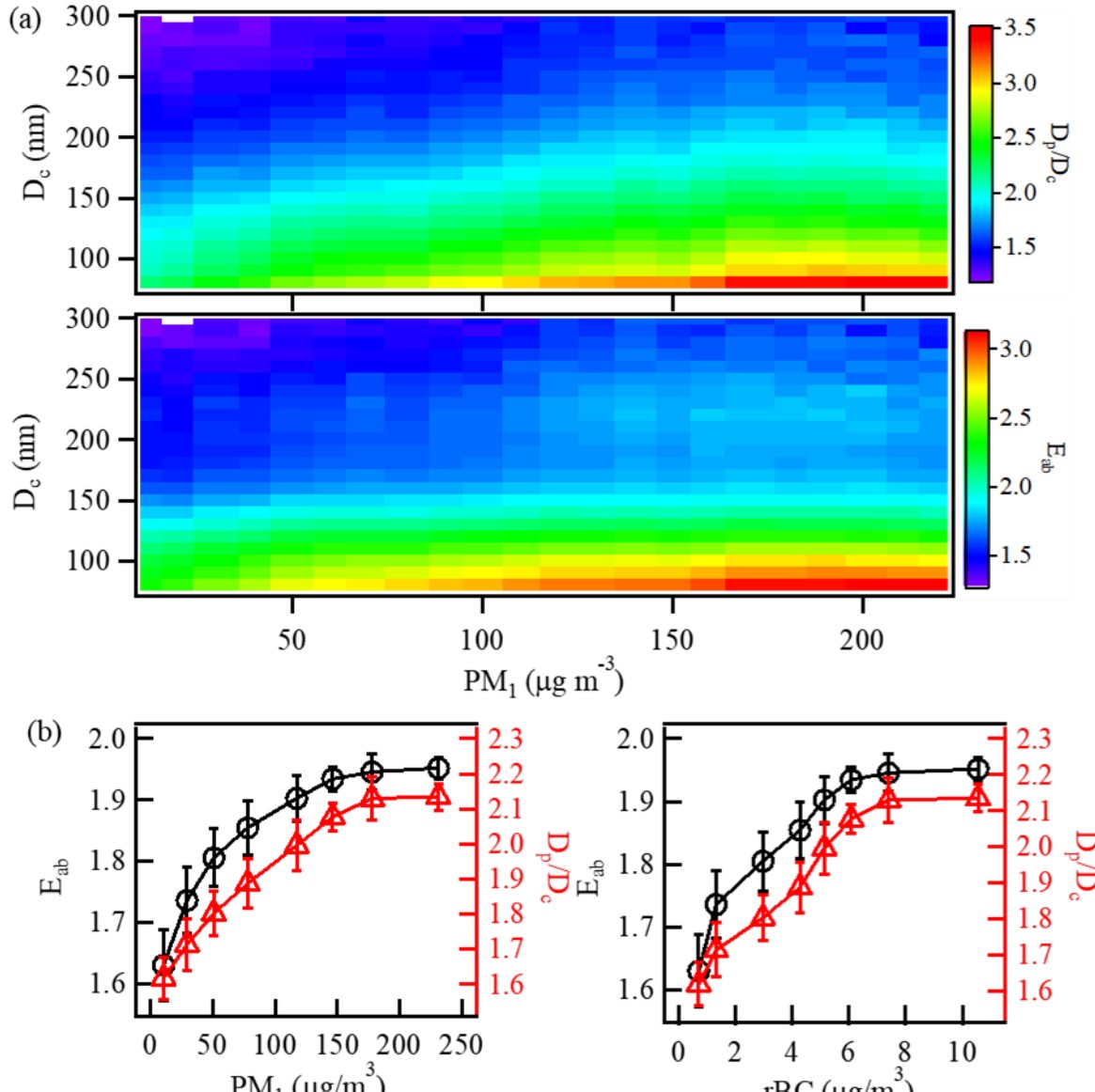

2 **Figure 2.** (a) The aging degree ($D_p/D_c$ ratio) and light absorption enhancement (calculated $E_{ab}$) of BC-containing particles with size-resolved
3 rBC cores ($D_c$) under different $PM_1$ concentration; (b) variations in the mass-averaged values of the $D_p/D_c$ ratio and calculated $E_{ab}$ of BC-
4 containing particles with $PM_1$ and rBC concentrations.

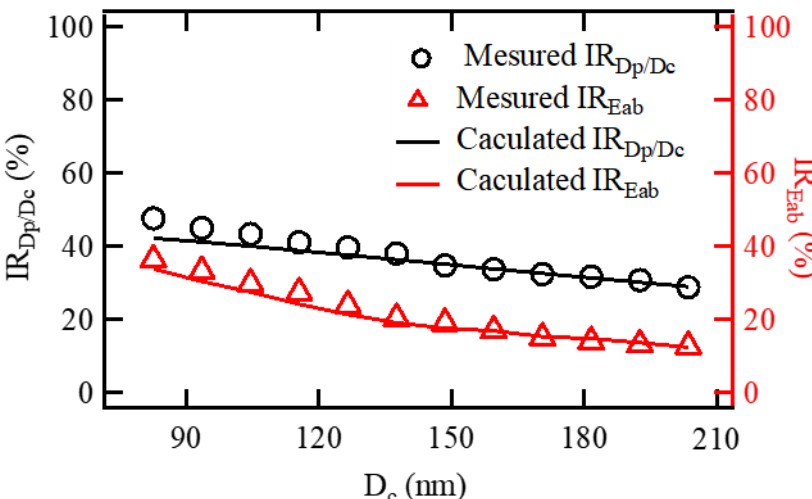

2 **Figure 3.** The increase ratio of the $D_p/D_c$ and and calculated $E_{ab}$ ($IR_{Dp/Dc}$ and $IR_{Eab}$) for BC-containing particles with PM$_1$

3 concentration increasing from 10 μg m$^{-3}$ to 230 μg m$^{-3}$. The calculated $IR_{Dp/Dc}$ and $IR_{Eab}$ values were determined based on Eqs.

4 (13) and (14). The measured $IR_{Dp/Dc}$ and $IR_{Eab}$ values were obtained from SP2 measurements.

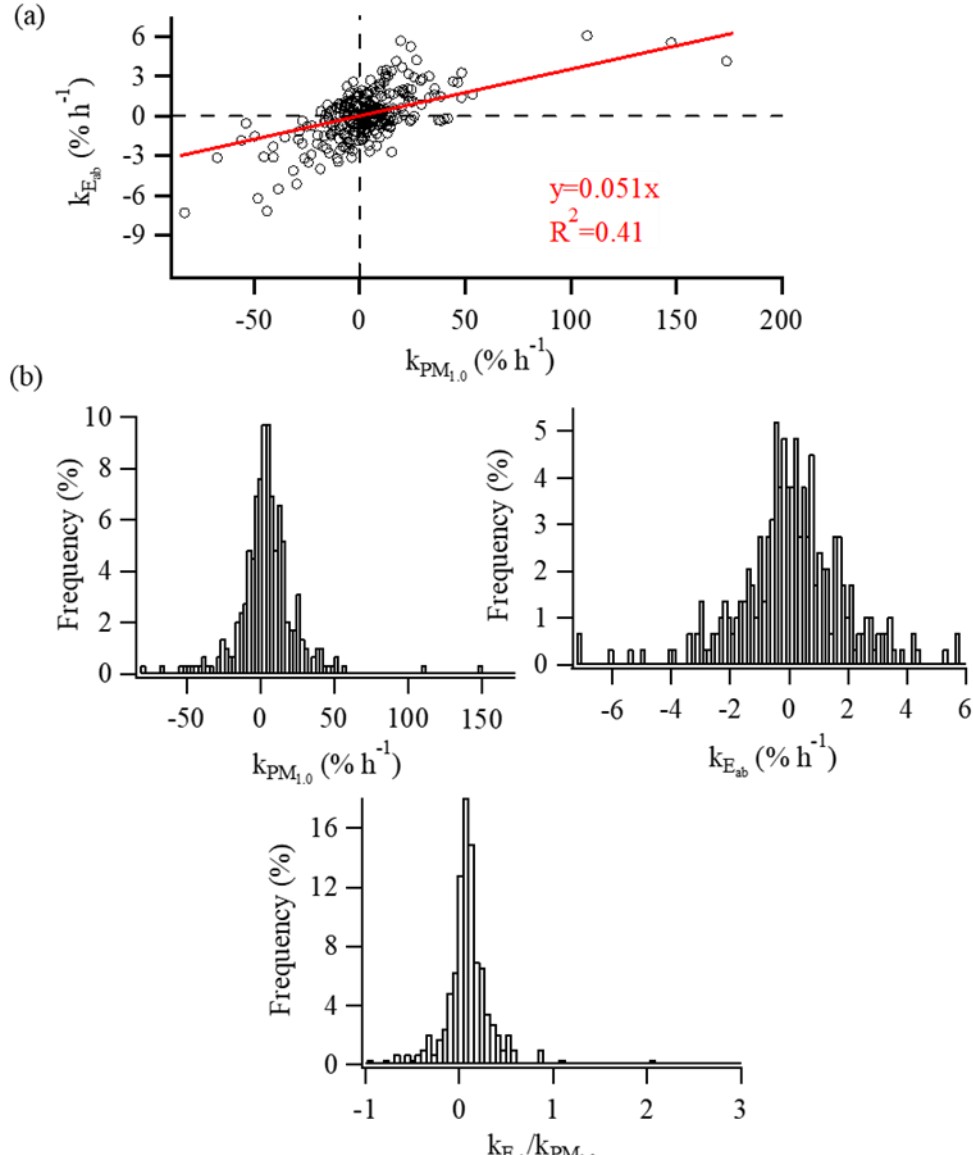

Figure 4. (a) Correlation between the changing rate of calculated $E_{ab}$ ($k_{Eab}$) and the changing rates of PM$_1$ concentrations ($k_{PM1}$) during the
campaign period. (b) Frequency distribution of $k_{Eab}$, $k_{PM1}$, and $k_{Eab}/k_{PM1}$. The $k_{Eab}$ and $k_{PM1}$ values represent an apparent changing rate of
calculated $E_{ab}$, PM$_1$ concentration and rBC mass concentration, respectively, and are from point-by-point differences of hourly $E_{ab}$, namely
$k_{Eab} = (E_{ab,t2}-E_{ab,t1})/(E_{ab,t1}*(t_2-t_1))$ and $k_{PM1} = (PM_{1,t2}- PM_{1,t1})/(PM_{1,t1}*(t_2-t_1))$. The sensitivity of the change of calculated $E_{ab}$ with changing
PM$_1$ concentrations was obtained by plotting $k_{Eab}$ vs. $k_{PM1}$ (i.e., $k_{Eab}/k_{PM1}$, the slope shown in (a)).

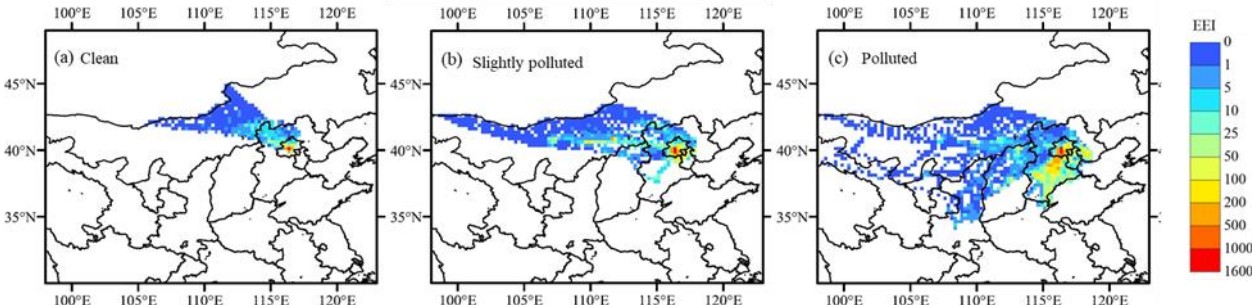

Figure 5. Spatial distribution (0.25°×0.25°) of the effective emission intensity (EEI, unit of ton/grid/year) for BC transported
to the observation site (40º00'17" N, 116º19'34" E) during clean, slightly polluted and polluted periods. The clean (PM$_{2.5}$ ≤
35 μg m$^{-3}$), slightly polluted (35 μg m$^{-3}$ < PM$_{2.5}$ ≤ 115 μg m$^{-3}$) and polluted (PM$_{2.5}$ > 115 μg m$^{-3}$) periods were classified
according to the Air Quality Index (Zheng et al. 2015). The site location and the boundaries of the in-region (i.e., Beijing) vs.
outside of region (i.e., other areas such as Tianjin, Heibei, Inner Mongolia, Shanxi, Shandong), shown in Fig. S1. Noted that
these regions are defined based on political boundaries.

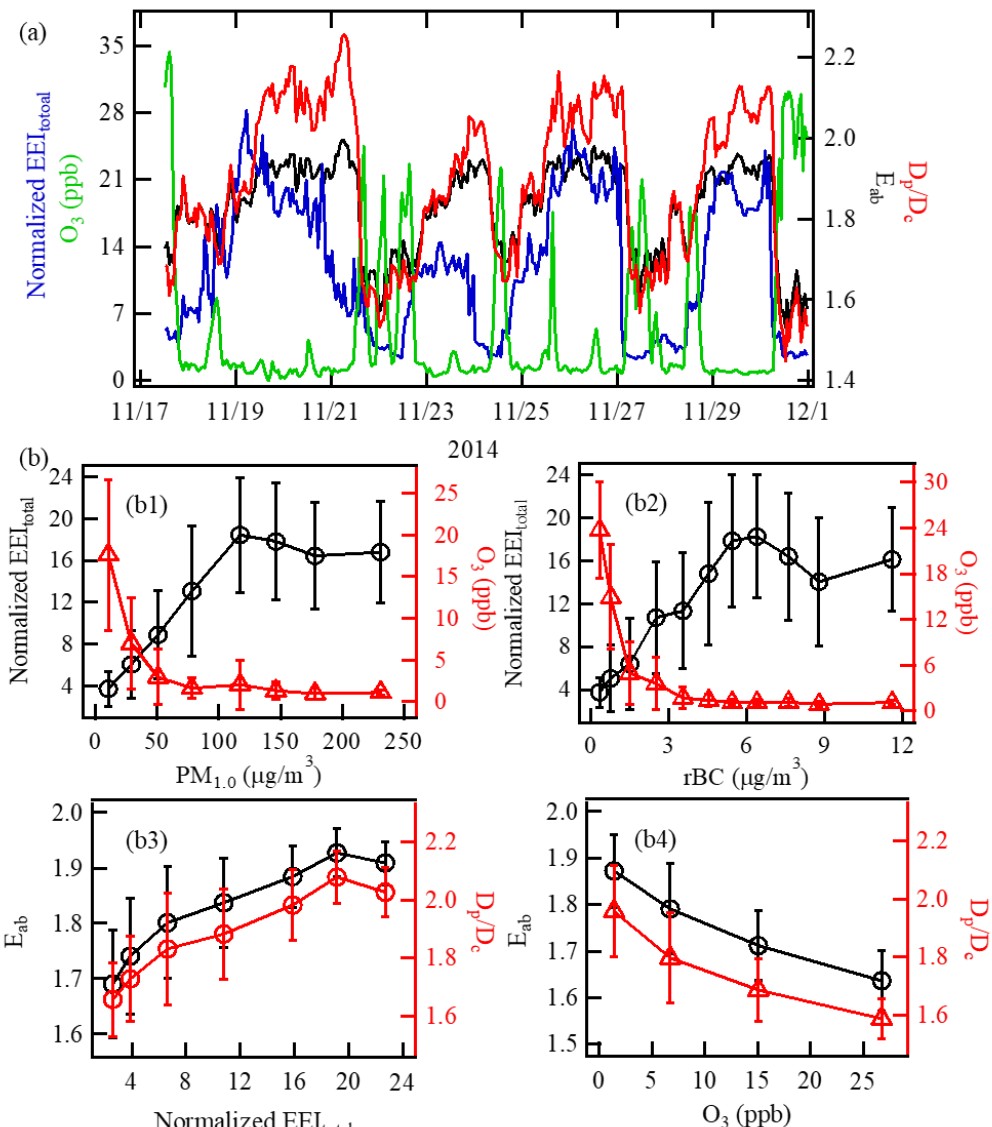

Figure 56. (a) Time series of the normalized $EEI_{total}$, $D_p/D_c$ rations and $E_{ab}$ of BC-containing particles and the O₃ concentration

during the campaign period. (b) Variations in the $D_p/D_c$ and $E_{ab}$ of BC-containing particles with the normalized $EEI_{total}$ and O₃

concentration. Normalized $EEI_{total}$ ($EEI_{total,normalized}$) was calculated by scaling by a factor of $10^{-3}$, namely $EEI_{total,normalized} =$

$EEI_{total}/1000$.

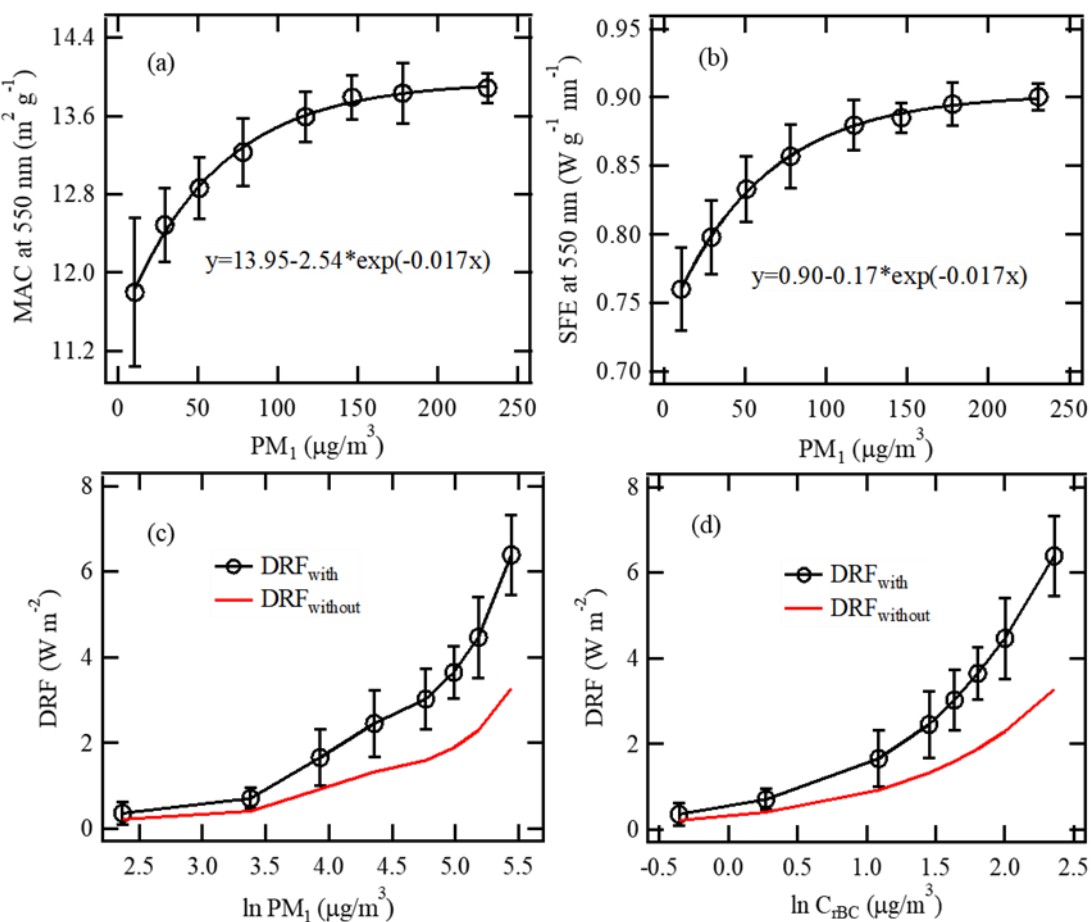

**Figure 7.** Variations in *MAC*, *SFE* and *DRF* and the *DRF*/rBC ratio of BC-containing particles under different pollution levels.
The DRF values for BC-containing particles were obtained by scaling the average *DRF* (0.31 W m$^{-2}$, Table S1) of externally
mixed BC from various climate models (Bond et al., 2013). The DRF$_{with}$ and DRF$_{without}$ values represent *DRF* with/without,
respectively, considering the differences of light-absorption capability of ambient BC-containing particles under different pollution levels;
namely, the DRF$_{with}$ was calculated with scaling factors of the $E_{ab}$ and rBC mass concentration ($C_{rBC}$) and the DRF$_{without}$ was
calculated with a scaling factor of $C_{rBC}$. Variations in DRF$_{with}$ under different pollution levels was due to the change of both the
light-absorption capability and mass concentration of BC; variations in *DRF*$_{without}$ was only attributed to the change of rBC
mass concentrations.

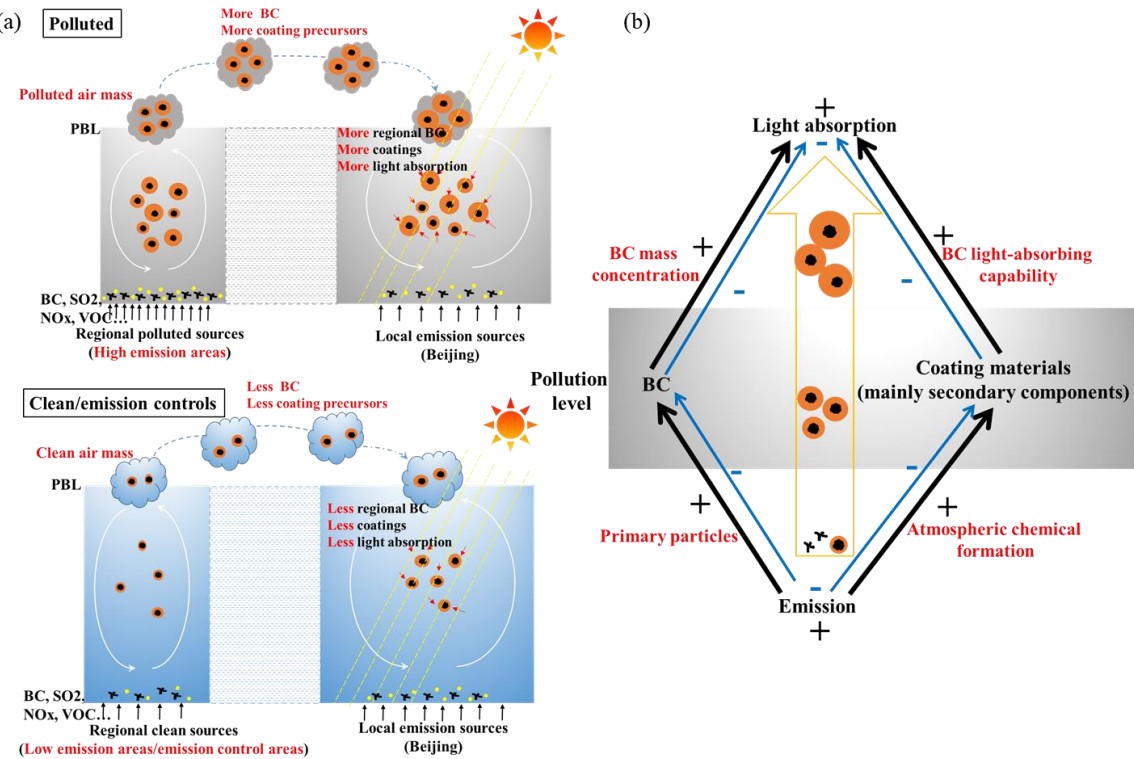

2 **Figure 8.** Conceptual scheme of amplification effect on BC light absorption associated with air pollution.