# Peer review of "Amplification of light absorption of black carbon associated with air"

_Atmospheric Chemistry and Physics, 2017_

## Referee Comment (RC1) · Anonymous Referee #1 · 16 Feb 2018

In this study, the authors examined light absorption of black carbon (BC) under clean and polluted conditions based on observations. They found that we found that the aging degree and light absorption capability of BC containing particles increased by 26-73% and 13-44% respectively, due to more coating materials on the BC surface. This work is interesting and merits publication after following comments addressed.

General Comments:

The authors reported a large amount of BC was originated from sources outside Beijing based on effective emission intensity. It is true in this analysis. But the authors need to caution that they were comparing the contributions from a small region (Beijing) and a large region (outside Beijing or adjacent regions). In addition, the authors evaluated the contribution of local photochemical production by the changes of O3 concentrations

in the atmosphere. They found that the O3 concentrations showed a different temporal trend. It only means weak photochemical production of O3 due to high aerosol concentrations blocking sunlight. It does not mean the local aging of BC is weak because high concentrations of aerosols may compensate the adverse conditions for BC aging. Anyway, the authors should provide uncertainty values for the numbers.

Specific comments:

Page 1 Line 14: It 'is' well known.

Page 2 Line 22: What is the 'lens effect'?

Page 7 Line 27: Missing ')'.

Page 8 Line 21: How many samples are there for different PM1 conditions?

Page 9 Line 27: It should be 'Figure 4'. The unit of EEI is 't/grid/year' shown in the figure. What does the 't' stand for?

Page 9 Line 30: 'account for' what? Does that mean the rest of the contribution is from Beijing's own emissions or emissions from other non-adjacent regions? As I understand, there are three emission source regions, emissions from Beijing, adjacent regions, and other regions. Please clarify.

Page 10 Line 1: Does EEItotal include the contribution from Beijing's own emissions?

Page 10 Line 3: I am confused that EEItotal increased by a factor of 4.6, but after that, the authors said BC from adjacent area. Should it be EEIadjacent? In addition, I think it needs a supplement to the conclusion that the increased BC is due to transport of polluted air mass, not the adverse local meteorology. It is true for a very small region, based on the analysis of this study, because the authors were comparing the contributions from a small region and a large region. Also, polluted events always occur over a larger region, even spread the whole eastern China. They are definitely caused by adverse meteorology. The more transport of pollutants into Beijing is probably a

consequence of increased pollutants due to adverse meteorology in other regions. For example, Yang et al. (2017) analyzed the source-receptor relationship of BC in China and found that, during polluted days in winter, the increases in BC over the North China Plain (including Beijing) is dominated by its local emissions instead of regions outside North China Plain. The weakening of winds can explain it.

Page 11 Line 1: What does the normalized EEI mean? Is it a percent value or some index?

Page 11 Line 19: I see the author calculated DRF by scaling the average DRF (0.32 W m-2) of externally mixed BC with an average MAC of 7.5 m2 g-1 from various climate models (Bond et al., 2013). Is the DRF value global average with a fixed BC climatology? The author should make it clear, or the readers may think the value is the DRF over Beijing during the analyzed clean and polluted days.

Page 12 Line 15: Delete 'by'.

Page 13 Line 19: It was defined as transport-controlled 'period'.

References

Yang, Y., Wang, H., Smith, S. J., Ma, P.-L., and Rasch, P. J.: Source attribution of black carbon and its direct radiative forcing in China, Atmos. Chem. Phys., 17, 4319-4336, https://doi.org/10.5194/acp-17-4319-2017, 2017.
* * *

---

## Referee Comment (RC2) · Anonymous Referee #3 · 17 Feb 2018

The authors investigated the evolution of BC optical properties, and concluded that under more polluted conditions, the aging process will enhance the coating of BC-containing particles and thus contribute to larger enhancement of BC particle light absorption. They further claim that pollution control strategy will have co-benefit effects on both air quality and climate. The content is suitable for publication within the scope of ACP, while some revisions are required. Please see detailed comments below.

Major issues:

1) The manuscript is still in need of a better discussion on uncertainties. Some examples are listed below, while I would suggest the authors do a systematic discussion on all the associated uncertainties, not just here and there.

   a. Page 4, lines 1-3: The authors mentioned the correction to Aethalometer data in the SI, where there is something confusing to me. First, the authors said they retrieved the correction factor by comparing absorption coefficients measured by AE and MAAP, but since AE was measuring at 660 nm while MAAP at 670 nm, are the authors just neglecting the difference? Second, the authors used an average value of 2.6 for all their AE measurements while they did determine a pretty wide range of the C, from 1.9 to 4.0, then how did the authors decide the uncertainty of 10% confidently?

   b. Page 4, line 26: the authors used 1.50-0i as the RIs value, is there any reason why? Is there some information on, e.g., the chemical compositions of the coating materials, to support that the use of 1.50-0i is reasonable? Otherwise I would suggest the authors consider some sensitivity test on RIs values as well as on RIc.

   c. Page 8, line 3: I am not sure how the authors determine that the Mie calculation has an uncertainty "smaller than 7%". The authors have shown in Figure S3 and the associated discussions that different RI values could result in 3%-10% difference in Dc. Assume it is on average 5%, then the mass concentration of rBC would be different by 16% (1.05^3, the cubic is converting from diameter to volume), not mentioning the uncertainties on the estimation of e.g., density, mixing, etc. I would suggest the authors do a much more careful job when they are talking about uncertainties.

2) About the processes contributing to the enhancement of BC light absorption. The authors are trying to add some discussions on the causes of BC coating and thus light absorption enhancement, but these discussions read somewhat weird if there is no sufficient evidence to support. Similarly, a couple of examples below:

   a. Page 8, line 18: "due to more secondary component formation", is it possible that more primary components were also emitted under the more polluted condition and coated onto the BC core during the aging process?

b. Page 10, lines 19-28: I do not understand why the authors are looking at the temporal trend of O3 to evaluate local photochemical processes. The trend of O3 could mean weak production, could mean strong ozonolysis (which could be dark reaction, i.e., nothing with photochemistry), or could just mean cloudy days thus no sunlight. This is not a sound reason for "weak local aging".

Minor issues:

The authors sometimes used "BC-containing particles" while sometimes "BC particles" and "BC" to name the same term, the BC-cored and other materials coated particles. Please try to be consistent throughout the manuscript, otherwise it will be confusing, e.g., Page 2, the "BC" of line 12, and the first "BC" of line 22, they did not actually have the same meaning.

Page 1, line 14: It "is" well known…

Page 2, line 5: both emissions of BC and the coating materials -> emissions of both BC and the coating materials;

Page 2, line 22: lens effect -> lensing effect. Same problem throughout the paper, e.g., Page 5, line 24, and Page 8, line 10, etc.

Page 2, line 23: results -> result;

Page 3, line 25: the particles were not "collected" by the diffusion dryer, please correct;

Page 4, line 25: not "RIs and RIc", here it should be RIs only.

Page 5, line 10: underestimate -> underestimation;

Page 5, equation (4) and equation (6): what is the difference between $m_{rBC}$ and $C_{rBC}$?

Page 7, line 21: that -> those;

Page 7, lines 22-24: The logic of this sentence is not 100% correct. Dp increases, which could be the increase of either Dc or coating materials, or both. The authors mentioned "simultaneous increase in the rBC mass concentration" exactly in the following sentence, which makes this sentence reads really weird. Same problem applies to the texts following Figure S8, that the authors only suggested the "18-fold" increase of $\sigma_{ab}$, and will need to provide more evidence on the "simultaneous increase" in both rBC and the coating materials.

Page 8, line 30: change rates -> changing rates;

Page 9, line 3: at our study -> in our study;

Page 9, line 21: (A) BC aging and (B) BC internally mixed with other components, it is hard to say A is the consequence of B, or vice versa;

Page 12, line 3: capacity or capability? Please note this is not the only place.

Page 12, lines 14-15: decreased by significantly -> decreased significantly;

Table 1, the unit is "$\mu g\ m^{-3}$", not "$\mu g\ cm^{-3}$";

Figure 2: Eab is not light absorption capability, it should be enhancement;

SI: page 4, line 13: what is "larges of coating materials"?

---

## Referee Comment (RC3) · Anonymous Referee #2 · 23 Mar 2018

The authors present measurements from Beijing, focusing on analysis and interpretation of data from a single particle soot photometer. The use the SP2 measurements to infer light absorption and light absorption amplification factors. The technical analysis is of reasonably high quality. The interpretation and discussion could benefit from some stronger quantitative analysis to discern process-based information. The authors also need to clarify how they have calculated averages, and whether they are mass-weighted or not. My specific comments follow below.

General comment on terminology: Throughout the manuscript, the authors need to clarify when they are talking about actual absorption measurements or absorption enhancement measurement, and when they are talking about calculated/theoretical values. There are many, many points where this distinction needs to be made clearly,

[Figure]

starting from the abstract and continuing through the conclusions. I will note only a few points where this is necessary as examples, but there are many more beyond what I have noted.

P1 L20 & 27, and P2 L1 & L3: This should be changed to "theoretical light absorption capability"

P2, L14: The Moffett study does not directly measure light absorption. Their conclusions are based on theoretical calculations. Thus, it is not appropriate here. Same for the Zhang et al. 2016 reference.

P2, L15: It is unclear as written when the authors are referring to theoretical studies versus observations studies. This must be clarified.

P2, L23: This should be revised to clarify that the concept of "more coating materials results in stronger light absorption capability" depends on whether one considers coatings on individual particles versus coatings averaged over the ensemble of particles.

P2 ,L29: It should be clarified that the idea that more materials coat BC under polluted conditions is only true so far as the total amount of BC does not scale equally with the overall amount of pollution. If there were more secondary aerosol but also more BC particles, then it is possible that the average coating per particle is unchanged in more versus less polluted conditions. Also, this statement oversimplifies issues related to mixing state, and whether that secondary material condenses on BC versus on non-BC containing particles. The authors oversimplify here, in my opinion.

P2, L31: it is unclear what "quasi-atmospheric" means. Also, this study ultimately indicates very simply that when you have greater amounts of coating on monodisperse particles the absorption enhancement is larger.

P4, L3: Fig. S2 indicates that the uncertainty in the aethelometer measurements is 10% based on the uncertainty in the "compensation factors." However, this inherently assumes that the measurements by the reference instrument, a MAAP, are perfect and

without uncertainty, which is not true. The actual uncertainty is larger and this should be noted. Also, this assumes that the MAAP perfectly accounts for filter-loading and multiple-scattering effects.

Eqn. 1: This is not so much an equation as a relationship. It does not seem to me that it needs to be called out as an equation.

SP2 limits: The authors indicate a lower size limit of 75 nm in the main text. But, Fig. S4 makes clear that the lowest two size bins are strongly biased low in terms of their concentrations since there is no physical reason for such a sharp fall off in concentration below ∼95 nm. It is unclear whether this is taken into account, which would be particularly important when the [PM1] is < 50 ug/m3. Fig. S4 makes clear that there is a counting artifact in the SP below 95 nm, where the detection efficiency falls off rapidly giving rise to an apparent sharp decrease in concentration.

Fig. S7 and P6/L5: it is unclear how this figure addresses uncertainties in the absorption calculations. However, the authors do argue in the supplemental that the "absorption. . .was underestimated by no more than 50%." A 50% uncertainty is very large and this information should not be buried in the supplemental. Further, additional details are required as to how this was determined. The MAC from Mie theory varies as a function of particle size while that from RDG is constant. The RDG MAC for bare BC, using the RI given here, is ∼3.2 m2/g at 880 nm. This is relatively small to begin with, so how is the 50% number determined. Also, the argument that "the uncertainty of BC light absorption from the calculation of bare BC properties using Mie theory is no more than 2%" is demonstrably not correct. If the absolute absorption can be underestimated by 50%, then the uncertainty cannot be only 2%. Just because there were few times that bare BC particles were observed does not change this fundamental issue. The uncertainty in the absolute absorption from the calculations is much larger than 2%. And then on P8, L5 it is stated that the uncertainty is 10%. It is ultimately unclear what the actual uncertainty on the calculations is.

P7/L19: This should be Fig. S4a.

P7/L26: This should say "calculated absorption coefficient."

P8/L2: As noted above, the uncertainty on the AE33 measurements is >10%.

P8/L5: is the agreement similarly good at the other wavelengths?

P8/L9: It should be clarified that these are all theoretical studies of the enhancement. None of the citations is to a direct measurement of the enhancement.

P8/L14: It should be clarified to "the calculated Eab".

P8/L16: It would be helpful to show a graph that explicitly has the Dp/Dc and calculated Eabs as a function of PM1 concentration, to help illustrate this point. This could be shown as a mean across all sizes, a weighted mean across all sizes, or for a few select sizes. This would help to illustrate the magnitude of the changes.

P8/L24: I find the term "aging degree" to be ambiguous because it could apply to almost anything. I suggest that here, and throughout, the authors change to a more specifically descriptive language. Perhaps "coating-to-core ratio"?

P8/L25: It would be helpful to frame this in the context of the overall size distribution, i.e. to report the weighted-average values based on the observed PM1-dependent BC core size distributions.

P8/L23: Better as "exponential function with a y-offset". However, this by itself gives little physical insight. In Metcalf et al. (2013), for example, there was a comparison to the expected decay based on the SA/V ratio of the particles and diffusion controlled growth. Here, the authors mention this study but do not connect to it quantitatively. The authors should strongly consider introducing a physical explanation using a semi-quantitative analysis, rather than just an empirical fit. I believe this would strengthen the paper.

P8/L30: it is unclear how the "change rates" were calculated. Are these point-by-point

differences? And, how is the Eab calculated? Is this a weighted average? Also, it's not entirely demonstrated how this is an especially meaningful metric. Isn't the same general information obtained by plotting Eab vs. rBC (for example)? Assuming this is from point-by-point differences, then one would expect that:

$$k\_Eab/k\_pm1 = ((E_{(ab,t2)}-E_{(ab,t1)})/E_{(ab,t1)})/((PM1\_t2-PM1\_t1)/(PM\_t1)) = (PM1\_t1)/E_{(ab,t1)} \cdot (E_{(ab,t2)}-E_{(ab,t1)})/(PM1\_t2-PM1\_t1)$$

How is this generally useful? This could be elaborated upon. Also, given that negative values are allowed, it could be clarified that these are not just "growth" rates. This is really just a susceptibility curve.

P8/L31: The units on the equation are incorrect. It is keab = 4.8kPM1, not 4.8%. The percents cancel.

P9/L1: The statement here relates to one point on a graph of hundreds of points. What is the uncertainty on a single point? Is this meaningful to state? I question whether it is especially meaningful to state the results for this one point. I could randomly pick another point, based on the maximum kPM1 (for example) and conclude that kEab varies slowly with kPM1. This feels to me too selective to be meaningful to include and I suggest it is removed or put in a fuller context.

P9/L2: The authors relate their observations to other studies. However, I do not understand why they only consider values with kEab > 0. Why exclude the negative numbers? Also, to reiterate my above point, are the individual points truly meaningful once one accounts for the uncertainty in the individual points? Using confidence intervals for the slopes would, in my opinion, be more meaningful. Or looking at the distribution of kEab values. It is evident from Fig. 3 that if a histogram of kEab values was made the peak would be around zero, i.e. that the particles are shrinking as often as they are growing, on average. While I do see some value in providing the range of values here, a much more statistical picture would provide much greater value.

Fig. S9 and P9/L5: If the authors were to introduce a more quantitative picture that included an interpretation of why this type of behavior might be expected that would be most welcome. Most likely, this is simply because the net change in diameter for a given amount of material deposited decreases with the size of the particle due to surface-to-volume scaling. Since the particles are larger when PM1 is larger, one would expect the deltaDp/time to decrease with PM1 and thus the Eab/time would also decrease. Of course, this oversimplifies because Eab is not a linear function of deltaDp. But, it would be great if the authors could introduce some physical discussion of why this observed behavior is/is not expected this would increase the value of this observation. P9/L9: I find this discussion about previous studies "ignoring" an aspect to be unclear and suggest it be expanded/clarified. How does the current observation help, specifically, explain these previous studies? It is not abundantly clear. Consider that the variability in the calculated Eabs is actually only ∼15% between the low and high PM1 periods in the size range that matters (the BC mass weighted size), based on Fig. 2. These previous measurements would not have been able to discern a 15% difference easily, most likely, in their data anyway. While the current study finds a large theoretical enhancement, what is not found is substantial variability in the enhancement (Fig. 2). The variability in the "growth rate" is inconsequential in the context of the actual enhancement dependence on PM1 (Fig. 2). Related, it is not clear where the 28% on L19 comes from. What matters is the mass-weighted enhancement. Assuming this is a straight average over the points in Fig. 2, this is not relevant to the actual measurement of the enhancement, which is weighted. I suggest that the authors rethink this discussion entirely.

Related to the previous comment, it is unclear how the references on P9/L9 relate to the references on P9/L13. The authors appear to be linking these, I think, but it is not clear. Also, this seems selective, as there are studies (e.g. Liu et al. 2015) in which variability was observed. The authors should aim to provide a more comprehensive picture. Finally, it is not at all clear that the conditions in the cited studies are similar enough to those here to be relevant. This aspect needs to be discussed.

P9/L27: Should be Figure 4.

Figure 4: The site location should be clearly indicated. In addition, the boundaries of the in-region vs. out of region should be indicated clearly. Also, it is not clear whether these regions are defined based on some physical parameter (e.g. as air basins) or simply based on political boundaries. It would be useful if this were addressed.

P10/L3: It is unclear where the 62% number comes from.

P10/L14: It is unclear how Fig. 4c indicates that there were "higher aging rates." Can this be clarified?

P10/L15: "observed" should be "calculated." Also, is this a weighted average? A straight average across size? This needs to be clarified. A weighted average is most appropriate. This comment applies to everywhere in the manuscript that values for Eab or Dp/Dc are mentioned. What sort of averages are these? This needs to be clarified.

P10: I find the discussion with respect to O3 is somewhat lacking in detail and nuance. While O3 is lower during the polluted events, the concentration of precursors may be higher and this would contribute to aging. Additionally, photochemical processing is not the only possible pathway. Have the authors considered to what extent NO3 oxidation at night might be important?

P10/L24: The meaning of "taking more EEItotal more BC" is unclear.

P11/L17: Is the MAC range given here the increase over the baseline or the actual MAC range at 550 nm? I find this unclear. Also, is this mass weighted? The MAC varies with particle size.

P11/L19: Are the DRF values given related to the total BC? It is surprising that the increase is so small, given that the BC concentration itself increased by a factor of 7 or so. This could be clarified.

P12/L5 and P12/L8: It is not clear to me that the authors have demonstrated that there

is a "speeding up" or "acceleration" of the coating process in the more polluted air. In fact, they seem to be arguing that photochemical processing is slower, but that there is longer time. This would actually go against the idea that there is a speeding up. This should be revisited. Associated with this, it is not clear to me that Fig. 7 is necessarily correct. The EEI analysis indicates that the contribution from the regional sources is smaller during less polluted periods. This does not mean that those particles from regional sources are less coated just because the overall particle distribution has less coating during low pollution periods. In fact, it is possible that the regional particles are more coated due to higher photochemical activity (potentially). But, because their fractional contribution is smaller the net impact on the coating amount appears smaller in the average, which is now dominated by the local sources. I think that Fig. 7 and the discussion section need to be rethought a little bit to provide a more nuanced picture of what might be happening. It may be that the authors are correct, but I do not think that they have fully justified their conclusion here.

P12, conclusions: The authors should consider reporting mass-weighted averages of Dp/Dc and Eab in addition to the ranges to provide a fuller picture.

P13/L6: See previous comment regarding the reporting of single points without stated uncertainties. Is the 7.3%/h value believable? It is unclear, since it is a single outlier in the entire plot.

P13/L10: It is unclear how a 13-44% variation in the Eabs translates to a 28% under-estimate in absorption. This appears to simply be an average of 13% and 44%, and not an appropriately mass-weighted average.

P13/L16: See previous comments about "speeding up". I do not think the authors have justified this conclusion.

P13/L23: It is unclear where the conclusion regarding heterogeneous chemistry comes from. This is pure speculation that is introduced at this point without justification. Why do the authors believe this to be the case? Also, this seems arbitrary. If the authors

had defined their periods differently then they could come to a different conclusion.

---

## Author Comment (AC1) · 15 Jun 2018

Anonymous Referee #1:

In this study, the authors examined light absorption of black carbon (BC) under clean and polluted conditions based on observations. They found that we found that the aging degree and light absorption capability of BC containing particles increased by 26-73% and 13-44% respectively, due to more coating materials on the BC surface. This work is interesting and merits publication after following comments addressed.

We would like to thank the reviewer for the valuable and constructive comments, which helps us to improve the manuscript. Listed below are our responses to the comments point-by-point, as well as the corresponding changes made to the revised manuscript. The reviewer's comments are marked in black and our answers are marked in blue, and the revision in the manuscript is further formatted as '*Italics*'.

**1. General Comments**

The authors reported a large amount of BC was originated from sources outside Beijing based on effective emission intensity. It is true in this analysis. But the authors need to caution that they were comparing the contributions from a small region (Beijing) and a large region (outside Beijing or adjacent regions). In addition, the authors evaluated the contribution of local photochemical production by the changes of O3 concentrations in the atmosphere. They found that the O3 concentrations showed a different temporal trend. It only means weak photochemical production of O3 due to high aerosol concentrations blocking sunlight. It does not mean the local aging of BC is weak because high concentrations of aerosols may compensate the adverse conditions for BC aging. Anyway, the authors should provide uncertainty values for the numbers.

**Response:** We thanks the reviewer for raising these questions and we hope the reviewer will be clear after our detailed clarification below.

(1) In terms of comparing the contributions of BC from a small region (Beijing) and a large region (outside Beijing or adjacent regions), we agree with the reviewer that we need to caution. Noted that we are not paying attention to the difference between the contributions of BC from local Beijing and other regions (For example, the contributions of BC from local Beijing (~37%) during polluted period was smaller than that from other regions (~63%), partly due to comparing the contributions from a small region (Beijing) and a large region (outside Beijing)). In this study, we focus on comparing the contributions of BC from outside Beijing (considered as regional origins in this study) among different pollution levels (i.e., clean, slight polluted and polluted period). During polluted period, we found that the BC amount from regional origins (i.e., other regions not including local Beijing) accounted for ~21%, ~39% and ~63% of total BC amount in the site during the clean, slightly polluted and polluted periods, respectively. This revealed that the regional contribution to BC over Beijing increased as the air pollution levels increased.

To make this point clear, the related discussion has been revised in the manuscript, as "*In this study, the spatial origin of total BC in the site was classified into local Beijing and other regions (i.e., adjacent areas, considered as regional origins in this study). Noted that the local region (i.e., Beijing) defined in this study is smaller than areas outside Beijing (e.g., Hebei, Tianjin, Shanxi and Inner Mongolia (Fig. S1)). Table 1 lists the contribution of BC from regional origins (i.e., $EEI_{ousiede}/EEI_{total}$ ratio). During polluted period, the contributions of BC from regional origins was ~63%, larger than that from local Beijing (~37%). This was partly due to comparing the contributions from a small region (Beijing) and a large region (outside Beijing). In this study, we focus on comparing the contributions of BC from outside Beijing (considered as regional origins in this study) among different pollution levels (i.e., clean, slight polluted and polluted period). The BC amount from regional origins (i.e., outside Beijing) accounted for ~21%, ~39% and ~63% of total BC amount in*

*the site during the clean, slightly polluted and polluted periods, respectively. This revealed that the regional contribution to BC over Beijing increased as the air pollution levels increased."*

(2) In terms of evaluating the contribution of local photochemical production by the changes of $O_3$ concentrations in the atmosphere, we have revised the related discussion, as *"When $PM_1$ concentrations were higher than ~120 μg $m^{-3}$, $O_3$ concentrations decreased to ~2 ppb. Zheng et al. (2015) has demonstrated the weakened importance of photochemistry in the production and aging of secondary aerosols in Beijing under polluted conditions due to decrease of oxidant concentrations. This indicated that the photochemical processing in BC aging may be weakened under higher polluted levels (i.e., $PM_1 > 120$μg $m^{-3}$). Noted that photochemical processing is not the only possible pathway in BC aging process and other pathways were not discussed in this study. The local aging process of BC might be enhanced by other pathways. For example, high concentrations of aerosols under polluted environment may compensate the adverse photochemical conditions for BC aging. "*

**2. Specific comments**

(1) Page 1 Line 14: It 'is' well known.

**Response:** Many thanks. We have revised it.

(2) Page 2 Line 22: What is the 'lens effect'?

**Response:** Thanks for the comment. We have stated/defined the "lens effect" in the revised manuscript according to the literatures (Bond et al., 2006; Fuller et al., 1999; Jacobson, 2001; Lack and Cappa, 2010): *"The non-BC species (i.e., coating materials) on the surface of BC can enhance BC light absorption via the lens effect (namely, the coating materials act as a lens to focus more photons on BC, Bond et al., 2006; Fuller*

*et al., 1999; Jacobson, 2001; Lack and Cappa, 2010)"*

(3) Page 7 Line 27: Missing ')'.

**Response:** Thanks. We apologize for the typo and have revised it.

(4) Page 8 Line 21: How many samples are there for different PM1 conditions?

**Response:** Thanks for the comments. In order to obtainthe evolution of $D_p/D_c$ ratio and $E_{ab}$ of BC-containing particles with size-resolved rBC cores with pollution development (shown in Fig. 2a), we used 28 different PM1 conditions.
To make this point clear, we revised the statement in the caption of Fig. 2a in the revised manuscript, as *"Figure 2. (a) The aging degree ($D_p/D_c$ ratio) and light absorption capability ($E_{ab}$) of BC-containing particles with size-resolved rBC cores ($D_c$) under different $PM_1$ concentration (28 samples)."*

(5) Page 9 Line 27: It should be 'Figure 4'. The unit of EEI is 't/grid/year' shown in the figure. What does the 't' stand for?

**Response:** Thanks. We apologize for the typo and have changed "Figure 5" in P9/L27 into *"Figure 4"*. The "t" stand for "ton", the unit of amount of air pollutant emission. We have changed "t/grid/year" into *"ton/grid/year"*.

(6) Page 9 Line 30: 'account for' what? Does that mean the rest of the contribution is from Beijing's own emissions or emissions from other non-adjacent regions? As I understand, there are three emission source regions, emissions from Beijing, adjacent regions, and other regions. Please clarify.

**Response:** Thanks to the reviewer to point this out. The "account for" here represents the proportion of BC amount from adjacent regions in total BC amount in the site. In this study, the spatial origin of total BC in the site was classified into local Beijing and other regions (e.g., Hebei, Tianjin, Shanxi and Inner Mongolia).

To make it clear, we have revised the manuscript, as "*In this study, the spatial origin of total BC in the site was classified into local Beijing and other regions (i.e., adjacent areas, considered as regional origins in this study). Noted that the local region (i.e., Beijing) defined in this study is smaller than areas outside Beijing (e.g., Hebei, Tianjin, Shanxi and Inner Mongolia (Fig. S1)). Table 1 lists the contribution of BC from regional origins (i.e., $EEI_{ousiede}/EEI_{total}$ ratio). During polluted period, the contributions of BC from regional origins was ~63%, larger than that from local Beijing (~37%). This was partly due to comparing the contributions from a small region (Beijing) and a large region (outside Beijing). In this study, we focus on comparing the contributions of BC from outside Beijing (considered as regional origins in this study) among different pollution levels (i.e., clean, slight polluted and polluted period). The BC amount from regional origins (i.e., outside Beijing) accounted for ~21%, ~39% and ~63% of total BC amount in the site during the clean, slightly polluted and polluted periods, respectively. This revealed that the regional contribution to BC over Beijing increased as the air pollution levels increased.*"

(7) Page 10 Line 1: Does EEItotal include the contribution from Beijing's own emissions?

**Response:** Thanks and yes. The $EEI_{total}$ includes the contribution from Beijing's own emissions, calculated by Eq. (8) in the manuscript.

(8) Page 10 Line 3: I am confused that EEItotal increased by a factor of 4.6, but after that, the authors said BC from adjacent area. Should it be EEIadjacent? In addition, I think it needs a supplement to the conclusion that the increased BC is due to transport of polluted air mass, not the adverse local meteorology. It is true for a very small region, based on the analysis of this study, because the authors were comparing the contributions from a small region and a large region. Also, polluted events always occur over a larger region, even spread the whole eastern China. They are definitely caused by adverse meteorology. The more transport of pollutants into Beijing is

probably a consequence of increased pollutants due to adverse meteorology in other regions. For example, Yang et al. (2017) analyzed the source-receptor relationship of BC in China and found that, during polluted days in winter, the increases in BC over the North China Plain (including Beijing) is dominated by its local emissions instead of regions outside North China Plain. The weakening of winds can explain it.

**Response:** Thanks for the comments. The EEI given here is the total EEI (EEI$_{total}$, calculated by Eqs. (7) and (8) in the manuscript) including BC from Beijing and other regions. The EEI$_{total}$ can be used to characterize the total BC amount (unit of ton/year, not the BC concentrations) transported to the site. The EEI$_{total}$ strongly depends on BC emission of source origins (including local Beijing and other regions) and dry/wet deposition during atmospheric transport. Considering the change of BC emission from local Bejing under different pollution levels was slight, the variations in EEI$_{total}$ was dominated by BC from different regional origins (i.e., higher EEI$_{total}$ due to BC from regional origins with higher emission (e.g., south of Hebei) and lower EEI$_{total}$ due to BC from regional origins with lower emission (e.g., Mongolia)). On the other hand, the effect of local meteorology on EEI$_{total}$ is slight. However, BC concentration in the site strongly depends on both total BC amount (transported from local Beijing and other regions, characterized by EEI$_{total}$ in this study) and local meteorology. In this study, we found that the EEI$_{total}$ and BC concentrations from the clean period to the polluted period increase by ~4.6 times and ~7.4 times, respectively, revealing that the increase of EEI$_{total}$ account for ~62% of the increase in BC mass concentration. This indicated that the adverse local meteorology contributed ~38% of the increase in BC mass concentration in the site from the clean period to the polluted period.

We agreed with the reviewer that less effect of adverse local meteorology is due to that local Bejing is smaller than other regions (e.g., Hebei, Tianjin, Shanxi and Inner Mongolia). Polluted events always occur over a larger region and are definitely caused by adverse meteorology. For our case, the adverse meteorology during polluted days in the whole large region including Beijing and other adjacent areas can lead to the increase of pollutants and then more transport of pollutants into Beijing.

Following the reviewer's suggestion, we have revised the statement to assess the effect of regional transport and adverse local meteorology on BC increase under polluted conditions, as "*Table 1 shows that the $EEI_{total}$ was 4.6 times higher during the polluted period than during the clean period, revealing that polluted air mass brought more BC to Beijing. BC concentration in the site strongly depends on both total BC amount (transported from local Beijing and other regions, characterized by $EEI_{total}$ in this study) and local meteorology. Table 1 shows that the BC concentrations from the clean period to the polluted period increase by ~7.4 times. The increase of $EEI_{total}$ (~4.6 times) accounted for ~62% the increase in BC mass concentrations (~7.4 times). This indicated that the adverse local meteorology contributed ~38% of the increase in BC mass concentration in the site from the clean period to the polluted period. Compared with regional transport, less effect of adverse local meteorology might be attributed to relatively small areas defined as the local region (i.e., Beijing) in this study. Polluted events in China always occur over a large region, e.g., North China Plain (Yang et al., 2017; Zheng et al., 2015). For our case, the adverse meteorology during polluted days in the whole large region including Beijing and other areas can lead to the increase of pollutants and then more transport of pollutants into Beijing. Yang et al. (2017) found that the increases in BC concentration under polluted conditions over the North China Plain (including Beijing and other adjacent areas) is dominated by its local emissions due to adverse meteorology.*"

(9) Page 11 Line 1: What does the normalized EEI mean? Is it a percent value or some index?

**Response:** Thanks to the reviewer to point this out. In this study the $EEI_{total}$ was normalized by scaling by a factor of $10^{-3}$, namely $EEI_{total,normalized} = EEI_{total}/1000$. To make it clear, we have added the statement in the caption in Fig.6 in the revised manuscript, as "*The normalized $EEI_{total}$ ($EEI_{total,normalized}$) was calculated by scaling by a factor of $10^{-3}$, namely $EEI_{total,normalized} = EEI_{total}/1000$.*"

(10) Page 11 Line 19: I see the author calculated DRF by scaling the average DRF (0.32 W m-2) of externally mixed BC with an average MAC of 7.5 m2 g-1 from various climate models (Bond et al., 2013). Is the DRF value global average with a fixed BC climatology? The author should make it clear, or the readers may think the value is the DRF over Beijing during the analyzed clean and polluted days.

**Response:** We thank the reviewer for raising the important issue. In this study, the DRF (0.31 W $m^{-2}$) of externally mixed BC was the global averages from the global climate models listed in Table R1 in the response (Table S1 in the revised manuscript). The calculated DRF of BC-containing particles (shown in the Fig.7 in the revised manuscript) was obtained by scaling the average $DRF$ (0.31 W $m^{-2}$) of externally mixed BC from various global climate models (Bond et al. 2013) with a scaling factor of $E_{ab}$ under different $PM_1$ concentrations. Noted that the DRF values calculated here did not consider the change of BC amount under different pollution levels. In this study, we focused on investigating the effect of BC light-absorption capability on DRF. Therefore, the increase in DRF of BC with increasing pollution levels just considered the change in light-absorption capability of BC.

Table R1 (Table S1 in the revised manuscript) The DRF of externally mixed BC from global climate models. The modeled values were taken from Bond et al. (2013).

| Global climate Model | Mixing state | Modeled MAC ($m^2$ $g^{-1}$) | Modeled DRF (W $m^{-2}$) | Reference |
|---|---|---|---|---|
| AeroCom models | | | | |
| GISS | External | 8.4 | 0.22 | Schulz et al. (2006) |
| LOA | External | 8.0 | 0.32 | Schulz et al. (2006) |
| LSCE | External | 4.4 | 0.30 | Schulz et al. (2006) |
| SPRINTARS | External | 9.8 | 0.32 | Schulz et al. (2006) |
| UIO-CTM | External | 7.2 | 0.22 | Schulz et al. (2006) |
| UMI | External | 6.8 | 0.25 | Schulz et al. (2006) |
| Other models | | | | |
| BCC_AGCM | External | 4.3 | 0.10 | Zhang et al. (2012) |
| CAM3 ECA | External | 10.6 | 0.57 | Kim et al. (2008) |
| GISS-GCM II | External | 7.8 | 0.51 | Chung and Seinfeld (2002) |
| Average values | | 7.5 | 0.31 | |

To make it clear, we added the Table S1 and the related discussion in the revised manuscript, as "*The DRF values for BC-containing particles at different pollution levels were obtained by scaling the average DRF (0.31 W m$^{-2}$) of externally mixed BC from various climate models (Bond et al. 2013) with a scaling factor of the calculated $E_{ab}$ under different PM$_1$ concentrations (Fig. 2b). The DRF (0.31 W m$^{-2}$) of externally mixed BC was the global averages from the global climate models listed in Table S1. In order to point out the effect of BC light-absorption capability on DRF under different PM1 concentrations, we did not consider the changes of BC amount for DRF calculation.*"

(11) Page 12 Line 15: Delete 'by'.

**Response:** Thanks. We have revised it.

(12) Page 13 Line 19: It was defined as transport-controlled 'period'.

**Response:** Thanks. We have changed "region" into "period".

**References:**

Bond, T. C., Doherty, S. J., Fahey, D. W., Forster, P. M., Berntsen, T., DeAngelo, B. J., Flanner, M. G., Ghan, S., Kärcher, B., Koch, D., Kinne, S., Kondo, Y., Quinn, P. K., Sarofim, M. C., Schultz, M. G., Schulz, M., Venkataraman, C., Zhang, H., Zhang, S., Bellouin, N., Guttikunda, S. K., Hopke, P. K., Jacobson, M. Z., Kaiser, J. W., Klimont, Z., Lohmann, U., Schwarz, J. P., Shindell, D., Storelvmo, T., Warren, S. G., and Zender, C. S.: Bounding the role of black carbon in the climate system: A scientific assessment, *J. Geophys. Res.-Atmos.*, 118, 5380-5552, 10.1002/jgrd.50171, 2013.

Bond, T. C., Habib, G., and Bergstrom, R. W.: Limitations in the enhancement of visible light absorption due to mixing state, J. Geophys. Res.-Atmos., 111, 2006.

Chung, S. H., and Seinfeld, J. H.: Global distribution and climate forcing of carbonaceous aerosols, *J. Geophys. Res.-Atmos.*, 107, AAC 14-11-AAC 14-33, 10.1029/2001JD001397, 2002.

Dongchul, K., Chien, W., L., E. A. M., C., B. M., and J., R. P.: Distribution and direct radiative forcing of carbonaceous and sulfate aerosols in an interactive size‐resolving aerosol‐climate model, *J. Geophys. Res.-Atmos.*, 113, doi:10.1029/2007JD009756, 2008.

Fuller, K. A., Malm, W. C., and Kreidenweis, S. M.: Effects of mixing on extinction by carbonaceous particles, *J. Geophys. Res.-Atmos.*, 104, 15941-15954, 1999.

Jacobson, M. Z.: Strong radiative heating due to the mixing state of black carbon in atmospheric aerosols, *Nature*, 409, 695-697, 2001.

Kim, D., Wang, C., Ekman, A. M. L., Barth, M. C., and Rasch, P. J.: Distribution and direct radiative forcing of carbonaceous and sulfate aerosols in an interactive size-resolving aerosol–climate model, *J. Geophys. Res.-Atmos.*, 113, 10.1029/2007JD009756, 2008.

Lack, D. A., and Cappa, C. D.: Impact of brown and clear carbon on light absorption enhancement, single scatter albedo and absorption wavelength dependence of black carbon, *Atmos. Chem. Phys.*, 10, 4207-4220, 2010.

Schulz, M., Textor, C., Kinne, S., Balkanski, Y., Bauer, S., Berntsen, T., Berglen, T., Boucher, O., Dentener, F., Guibert, S., Isaksen, I. S. A., Iversen, T., Koch, D., Kirkevåg, A., Liu, X., Montanaro, V., Myhre, G., Penner, J. E., Pitari, G., Reddy, S., Seland, Ø., Stier, P., and Takemura, T.: Radiative forcing by aerosols as derived from the AeroCom present-day and pre-industrial simulations, *Atmos. Chem. Phys.*, 6, 5225-5246, 2006.

Yang, Y., Wang, H., Smith, S. J., Ma, P.-L., and Rasch, P. J.: Source attribution of black carbon and its direct radiative forcing in China, *Atmos. Chem. Phys.*, 17, 4319-4336, 2017.

Zhang, H., Wang, Z., Wang, Z., Liu, Q., Gong, S., Zhang, X., Shen, Z., Lu, P., Wei, X., Che, H., and Li, L.: Simulation of direct radiative forcing of aerosols and their effects on East Asian climate using an interactive AGCM-aerosol coupled system, *Clim. Dyn.*, 38, 1675-1693, 10.1007/s00382-011-1131-0, 2012.

Zheng, G. J., Duan, F. K., Su, H., Ma, Y. L., Cheng, Y., Zheng, B., Zhang, Q., Huang, T., Kimoto, T., Chang, D., Pöschl, U., Cheng, Y. F., and He, K. B.: Exploring the severe winter haze in Beijing: the impact of synoptic weather, regional transport and heterogeneous reactions, *Atmos. Chem. Phys.*, 15, 2969-2983, 2015.

---

## Author Comment (AC2) · 15 Jun 2018

Anonymous Referee #3:

The authors investigated the evolution of BC optical properties, and concluded that under more polluted conditions, the aging process will enhance the coating of BC-containing particles and thus contribute to larger enhancement of BC particle light absorption. They further claim that pollution control strategy will have co-benefit effects on both air quality and climate. The content is suitable for publication within the scope of ACP, while some revisions are required. Please see detailed comments below.

We would like to thank the reviewer for the valuable and constructive comments, which helps us to improve the manuscript. Listed below are our responses to the comments point-by-point, as well as the corresponding changes made to the revised manuscript. The reviewer's comments are marked in black and our answers are marked in blue, and the revision in the manuscript is further formatted as '*Italics*'.

**1. Major issues**

(1) The manuscript is still in need of a better discussion on uncertainties. Some examples are listed below, while I would suggest the authors do a systematic discussion on all the associated uncertainties, not just here and there.
**Response:** We thank the reviewer for raising the important issue. Following the reviewer's suggestion, we have systematically discussed all the associated uncertainties. Here, to the uncertainties mentioned by the reviewer is as follows.

(a) Page 4, lines 1-3: The authors mentioned the correction to Aethalometer data in the SI, where there is something confusing to me. First, the authors said they retrieved the correction factor by comparing absorption coefficients measured by AE and MAAP, but since AE was measuring at 660 nm while MAAP at 670 nm,

are the authors just neglecting the difference? Second, the authors used an average value of 2.6 for all their AE measurements while they did determine a pretty wide range of the C, from 1.9 to 4.0, then how did the authors decide the uncertainty of 10% confidently?

**Response:** We thank the reviewer for raising this question. For the first question, we have discussed the uncertainty from the inconsistence of wavelength for AE and MAAP measurements. Considering that the absorption is inversely proportional to wavelength (*Bond and Bergstrom, 2006*), the difference of wavelength for AE (at 660 nm) and MAAP (at 670 nm) measurements, would lead to an uncertainty of ~1.5% for the corrected absorption coefficients in AE measurement. The related statement has been added in the revise supplement, as "*Noted that the AE and MAAP measurements used to calculate the factor C were at different wavelengths, namely 660 nm and 670 nm, respectively. Considering that the absorption is inversely proportional to wavelength (Bond and Bergstrom, 2006), the difference in wavelength would lead to an uncertainty of ~1.5% for the corrected absorption coefficients in AE measurement.*"

To the second question, we have reestimated the uncertainty related to the factor *C*. The uncertainty in the factor *C* of AE measurements obtained in our study was dominated by the uncertainty in MAAP measurements. In this study, we corrected the MAAP data using the algorithm reported by Hyvärinen et al. (2013). Hyvärinen et al. (2013) compared the results from a PAS against those derived from the MAAP in Beijing, and estimated the uncertainty of ~15% in absorption coefficients derived from MAAP based on the developed algorithm. Therefore, we estimated that the factor *C* derived by comparing AE and MAAP measurement would exhibit an uncertainty of ~15%. The related statement has been revised as "*In this study, the uncertainty in the factor C was dominated by the uncertainty in MAAP measurements. We corrected the MAAP data using the algorithm reported by Hyvärinen et al. (2013). They estimated that the uncertainty in absorption coefficients derived from MAAP based on the developed algorithm was ~15% by comparing the results from a PAS against those derived from the MAAP in Beijing.*

*This indicated that the factor C used in our study (~2.6) would exhibit an uncertainty of ~15% from the uncertainty in MAAP measurements.*"

(b) Page 4, line 26: the authors used 1.50-0i as the RIs value, is there any reason why? Is there some information on, e.g., the chemical compositions of the coating materials, to support that the use of 1.50-0i is reasonable? Otherwise I would suggest the authors consider some sensitivity test on RIs values as well as on RIc.

**Response:** Thanks for the comment. Following the reviewer's suggestion, we have demonstrated that the use of 1.50-0i for the refractive index of coating materials based on their chemical compositions during the campaign period (Fig. R1 in the response and new Fig. S4 in the revised manuscript). It is known from the literature (Schkolnik et al., 2007; Mallet et al., 2003; Marley et al., 2001) that major inorganic components of ambient aerosol from urban emission (nitrate, sulfate, mineral dust, sea salt and trace metal) have a refractory of (1.5-1.6)-0i and there is a range of (1.4-1.5)-0i for the refractory of organic components. In this study, we used the values of 1.55-0i and 1.45-0i as refractive indexes of inorganic and organic components of coating materials. The components of coating materials was similar to non-refractory compositions in $PM_1$ particles (Peng et al., 2016). Figure R1 (new Fig. S4 in the revised manuscript) shows that the fraction of inorganic and organic components in coating materials are ~51% and ~49%, respectively. The refractive index of a mixture particle can be calculated as the volume weighted average of the refractive indices of all components (Hänel, et al. 1968; Marley et al., 2001; Bond and Bergstrom, 2006; Schkolnik et al., 2007), as $\tilde{m} = \sum_i \tilde{m}_i c_i$, where $\tilde{m}$ is the refractive index of a mixture particle; $\tilde{m}_i$ is the refractive index of particle species; c is the volume ratio of particle species. Based on the equation, the refractive index of coating materials of BC-containing particles ($RI_s$) was ~1.50-0i during the campaign period.

On the other hand, we have considered some sensitivity test on the refractive index of rBC cores ($RI_c$), see the statement in page 4 line 26-28 in the manuscript and Fig S3 (new Fig S4 in the revised supplement) in the supplement.

[Figure]

Figure R1 (Fig. S4 in the revised manuscript). Non-refractory compositions of PM$_1$ particles during the campaign period.

To make this point clear, we have added Fig. S2 and the related discussion in the revised manuscript, as "*The RI$_s$ value used in this study are 1.50-0i based on the chemical compositions of coating materials during the campaign period. The components of coating materials was similar to non-refractory compositions in PM1 particles (Peng et al., 2016). Figure S4 reveals that the fraction of inorganic and organic components in coating materials of BC-containing particles are ~51% and ~49%, respectively. It is known from the literature (Schkolnik et al., 2007; Mallet et al., 2003; Marley et al., 2001) that major inorganic components of ambient aerosol from urban emission (nitrate, sulfate, mineral dust, sea salt and trace metal) have a refractory of (1.5-1.6)-0i and there is a range of (1.4-1.5)-0i for the refractory of organic components. In this study, we used the values of 1.55-0i and 1.45-0i as refractive indexes of inorganic and organic components of coating materials. The refractive index of a mixture particle can be calculated as the volume weighted average of the refractive indices of all components (Hänel, et al. 1968; Marley et al., 2001; Bond and Bergstrom, 2006; Schkolnik et al., 2007), as $\tilde{m} = \sum_i \tilde{m}_i c_i$, where $\tilde{m}$ is the refractive index of a mixture particle; $\tilde{m}_i$ is the refractive index of particle species; c is the volume ratio of particle species. Based*

*on the equation, the refractive index of coating materials of BC-containing particles ($RI_s$) was ~1.50-0i during the campaign period.* "

(c) Page 8, line 3: I am not sure how the authors determine that the Mie calculation has an uncertainty "smaller than 7%". The authors have shown in Figure S3 and the associated discussions that different RI values could result in 3%-10% difference in Dc. Assume it is on average 5%, then the mass concentration of rBC would be different by 16% (1.05^3, the cubic is converting from diameter to volume), not mentioning the uncertainties on the estimation of e.g., density, mixing, etc. I would suggest the authors do a much more careful job when they are talking about uncertainties.

**Response:** Thanks for the comments. There might be some misunderstanding on the uncertainty of Mie calculation given here. We would like to kindly clarify that the uncertainty of 3%-10% shown in Fig. S3 from different $RI_c$ values in Mie calculation was for the whole diameter of BC-containing particles ($D_p$), not for diameter of rBC core ($D_c$). We did not use Mie calculation to determine the mass concentration of rBC, which was obtained from SP2 measurements. Therefore, the mass concentration of rBC would not be different due to using different $RI_c$ values. We have recalculated the uncertainties on the calculated light absorption. In this study, the MAC for bare BC derived from Mie calculation, using the RI (i.e., 2.26-1.26i) given here, is 3.5-4.4 $m^2$/g at 880 nm (Fig. R2 in the response). Bond and Bergstrom (2006) suggested a value of 7.5 $m^2$/g for the MAC of bare BC at 550 nm. Considering that the absorption is inversely proportional to wavelength (Bond and Bergstrom, 2006), the MAC of bare BC at 880 nm is estimated to be ~4.7 $m^2$/g, which was slightly greater than that (~4.3 $m^2$/g) obtained from Mie calculation in our study. This indicated the uncertainty of MAC for bare BC from Mie calculation was ~8%. We estimated that the uncertainties of calculated BC light absorption related to MAC of bare rBC from Mie calculation was ~8%.

[Figure]

Figure R2 (Fig. S7 in the revised manuscript). The time series of MAC derived from Mie calculation for BC cores (i.e., bare BC) at 880 nm.

Correspondingly, we added the new Fig. S7 and related discussion in the revised supplement, as "*Based on Mie calculation, we obtained the MAC of rBC core ($MAC_c$) at 880 nm in the range of 3.8-4.5 $m^2/g$ with an average of ~4.3 $m^2/g$ during the campaign period (Fig. S9). Bond and Bergstrom (2006) suggested a value of 7.5 $m^2/g$ for the MAC of bare BC at 550 nm. Considering that the absorption is inversely proportional to wavelength (Bond and Bergstrom, 2006), the MAC of bare rBC at 880 nm is estimated to be ~4.7 $m^2/g$, which was slightly greater than that (~4.3 $m^2/g$) obtained from Mie calculation in our study. This indicated the uncertainty of MAC for bare rBC from Mie calculation was ~8%. We estimated that the uncertainties of calculated BC light absorption related to MAC of bare rBC from Mie calculation was ~8%.*"

(2) About the processes contributing to the enhancement of BC light absorption. The authors are trying to add some discussions on the causes of BC coating and thus light absorption enhancement, but these discussions read somewhat weird if there is no sufficient evidence to support. Similarly, a couple of examples below:

**Response:** Thanks for the comment. Following the reviewer's suggestion, we

have modified some discussion on the causes of BC coating and thus light absorption enhancement in the revised manuscript to make them more appropriate. Here, to the discussions/statements mentioned by the reviewer is as follows.

(a) Page 8, line 18: "due to more secondary component formation", is it possible that more primary components were also emitted under the more polluted condition and coated onto the BC core during the aging process?

**Response:** Thanks to the reviewer for raising this concern. We agree with the reviewer. The statement has been revised as "*In terms of BC-containing particles with a certain rBC core size, their $D_p/D_c$ ratio and $E_{ab}$ were greater under higher $PM_1$ concentrations, which could be attributed to more coating materials on BC surface under more pollution environment. The increase of both primary and secondary components under more polluted conditions was favorable to BC aging by coagulation and condensation, which happen mostly between BC and non-BC species.*"

(b) Page 10, lines 19-28: I do not understand why the authors are looking at the temporal trend of O3 to evaluate local photochemical processes. The trend of O3 could mean weak production, could mean strong ozonolysis (which could be dark reaction, i.e., nothing with photochemistry), or could just mean cloudy days thus no sunlight. This is not a sound reason for "weak local aging".

**Response:** Thanks to the reviewer to point this out. We agree with the reviewer that the decrease in O3 concentration can not fully support weaken local aging. In this study, we focused on the effect of regional transport on BC aging process. We just roughly discussed the chemical process during BC aging. The chemical process of BC aging under polluted environment in china is complex, which involved photochemical oxidation and heterogeneous chemical production (Zheng et al. 2015). We will investigate the chemical process of BC aging under polluted environment in future.

In the revised manuscript, we have toned down the related discussion on the

chemical process of BC aging, as "*When $PM_1$ concentrations were higher than ~120 μg $m^{-3}$, $O_3$ concentrations decreased to ~2 ppb. Zheng et al. (2015) has demonstrated the weakened importance of photochemistry in the production and aging of secondary aerosols in Beijing under polluted conditions due to decrease of oxidant concentrations. This indicated that the photochemical processing in BC aging may be weakened under higher polluted levels (i.e., $PM_1 > 120μg$ $m^{-3}$). Noted that photochemical processing is not the only possible pathway in BC aging process and other pathways were not discussed in this study. For example, high concentrations of aerosols under polluted environment may compensate the adverse photochemical conditions for BC aging.*"

**2. Minor issues**

(1) The authors sometimes used "BC-containing particles" while sometimes "BC particles" and "BC" to name the same term, the BC-cored and other materials coated particles. Please try to be consistent throughout the manuscript, otherwise it will be confusing, e.g., Page 2, the "BC" of line 12, and the first "BC" of line 22, they did not actually have the same meaning.
**Response:** Thanks for the comment. Throughout the manuscript, we have revised the terms to keep them consistent.

(2) Page 1, line 14: It "is" well known…
**Response:** Thanks. We have revised it.

(3) Page 2, line 5: both emissions of BC and the coating materials -> emissions of both BC and the coating materials;
**Response:** Thanks. We have revised it.

(4) Page 2, line 22: lens effect -> lensing effect. Same problem throughout the paper, e.g., Page 5, line 24, and Page 8, line 10, etc.

**Response:** Thanks. We have revised them.

(5) Page 2, line 23: results -> result;

**Response:** Thanks. We have revised it.

(6) Page 3, line 25: the particles were not "collected" by the diffusion dryer, please correct;

**Response:** Thanks for the comment. We have revised the sentence as "*Ambient aerosol particles were collected by a PM₁ cyclone and then passed through a diffusion silica gel dryer…..*"

(7) Page 4, line 25: not "RIs and RIc", here it should be RIs only.

**Response:** Thanks. We have revised it.

(8) Page 5, line 10: underestimate -> underestimation;

**Response:** Thanks. We have revised it.

(9) Page 5, equation (4) and equation (6): what is the difference between mrBC and CrBC?

**Response:** Thanks. The $m_{rBC}$ represents the mass of a single rBC core (see the statement in the page 4, line 21 in the manuscript), and the $C_{rBC}$ is the rBC mass concentration (see the statement in the page 6, line 3 in the manuscript).

(10) Page 7, line 21: that -> those;

**Response:** Thanks. We have revised it.

(11) Page 7, lines 22-24: The logic of this sentence is not 100% correct. Dp increases, which could be the increase of either Dc or coating materials, or both. The authors mentioned "simultaneous increase in the rBC mass concentration" exactly in the following sentence, which makes this sentence reads really weird. Same problem applies to the texts following Figure S8, that the authors only suggested the "18-fold" increase of σab, and will need to provide more evidence on the "simultaneous increase" in both rBC and the coating materials.

**Response:** Thanks. Following the reviewer's suggestion, we have added the Fig. R3 (new Fig. S8 in the revised manuscript) to support "simultaneous increase in rBC mass concentration and the amount of coating materials". Correspondingly, the sentences in   Page 7, lines 22-24 were revised as "*Moreover, the $D_p$ exhibited sustained growth from ~180 nm to ~400 nm during a pollution episode, which could be a consequence of the increase in either $D_c$ or coating materials, or both. Figure S6a shows a slight change in $D_c$ with pollution development. However, the coating thickness of BC-containing particles increased with PM1 concentration (Fig. S10a). Therefore, the sustained growth of $D_p$ during a pollution episode was dominated by more coating materials under more polluted conditions. Figure S10 shows the simultaneous increase in the rBC mass concentration and the amount of coating materials on the BC surface, which could significantly enhance the light absorption of BC-containing particles.*", and the statement following Fig. S8 (new Figure S11 in the revised supplement) in the revised supplement was revised as "*The simultaneous increase in the rBC mass concentration and the amount of coating materials shown in Fig. S10 revealed that the increase of $\sigma_{ab,880nm}$ (~18 fold from ~10 μg $m^{-3}$ of $PM_1$ to ~230 μg $m^{-3}$ of $PM_1$) could be attributed to simultaneous increase in the rBC mass concentration and the amount of coating materials on the BC surface.*"

[Figure]

Figure R3 (Fig. S10 in the revised manuscript). Variations in the coating thickness of BC-containing particles with the (a) PM$_1$ and (b) rBC mass concentrations.

(12) Page 8, line 30: change rates -> changing rates;

**Response:** Thanks. We have revised it.

(13) Page 9, line 3: at our study -> in our study;

**Response:** Thanks. We have revised it.

(14) Page 9, line 21: (A) BC aging and (B) BC internally mixed with other components, it is hard to say A is the consequence of B, or vice versa;

**Response:** Thanks. We have revised the sentence as "*BC aging in the atmosphere, namely BC internally mixing with other aerosol components, is associated with atmospheric transport (Gustafsson and Ramanathan, 2016).*"

(15) Page 12, line 3: capacity or capability? Please note this is not the only place.

**Response:** Thanks. Throughout the manuscript, we have changed "capacity" into "*capability*".

(16) Page 12, lines 14-15: decreased by significantly -> decreased significantly;

**Response:** Thanks. We have revised it.

(17) Table 1, the unit is "µg m-3", not "µg cm-3";

**Response:** Thanks. We have revised it.

(18) Figure 2: Eab is not light absorption capability, it should be enhancement;

**Response:** Thanks. We have revised it.

(19) SI: page 4, line 13: what is "larges of coating materials"?

**Response:** Thanks. We have changed "larges of coating materials" into "*significantly larger in volume of coating materials than that of rBC cores*".

**References:**

Bond, T. C., and Bergstrom, R. W.: Light Absorption by Carbonaceous Particles: An Investigative Review, *Aerosol Sci. Technol.*, 40, 27-67, 2006.

Gustafsson, Ö., and Ramanathan, V.: Convergence on climate warming by black carbon aerosols, *Proc. Natl. Acad. Sci.* USA, 113, 4243-4245, 2016.

Hänel, G.: The real part of the mean complex refractive index and the mean density of samples of atmospheric aerosol particles, *Tellus*, 20, 371-379, 1968.

Hyvärinen, A.-P., Vakkari, V., Laakso, L., Hooda, R. K., Sharma, V. P., Panwar, T. S., Beukes, J. P., van Zyl, P. G., Josipovic, M., Garland, R. M., Andreae, M. O., Pöschl, U., and Petzold, A.: Correction for a measurement artifact of the Multi-Angle Absorption Photometer (MAAP) at high black carbon mass concentration levels, *Atmos. Meas. Tech.*, 6, 81-90, 2013.

Peng, J., Hu, M., Guo, S., Du, Z., Zheng, J., Shang, D., Levy Zamora, M., Zeng, L., Shao, M., Wu, Y.-S., Zheng, J., Wang, Y., Glen, C. R., Collins, D. R., Molina, M. J.,

and Zhang, R.: Markedly enhanced absorption and direct radiative forcing of black carbon under polluted urban environments, *Proc. Natl. Acad. Sci.* USA,113, 4266-4271, 2016.

Schkolnik, G., D. Chand, A. Hoffer, M.O. Andreae, C. Erlick, E. Swietlicki and Y. Rudich (2007), Constraining the density and complex refractive index of elemental and organic carbon in biomass burning aerosol using optical and chemical measurements, *Atmos. Environ.*, 41, 1107-1118.

Mallet, M., J.C. Roger, S. Despiau, O. Dubovik and J.P. Putaud (2003), Microphysical and optical properties of aerosol particles in urban zone during ESCOMPTE. *Atmos. Res.*, 69(1-2), 73-97.

Marley, N.A., J.S. Gaffney, C. Baird, C.A. Blazer, P. J. Drayton and J. E. Frederick (2001), An empirical method for the determination of the complex refractive index of size fractionated atmospheric aerosols for radiative transfer calculations, *Aerospace. Sci. Technol.* 34(6), 535-549.

Zheng, G. J., Duan, F. K., Su, H., Ma, Y. L., Cheng, Y., Zheng, B., Zhang, Q., Huang, T., Kimoto, T., Chang, D., Pöschl, U., Cheng, Y. F., and He, K. B.: Exploring the severe winter haze in Beijing: the impact of synoptic weather, regional transport and heterogeneous reactions, *Atmos. Chem. Phys.*, 15, 2969-2983, 2015.

---

## Author Comment (AC3) · 15 Jun 2018

Anonymous Referee #2:

We would like to thank the reviewer for the valuable and constructive comments, which helps us to improve the manuscript. Listed below are our responses to the comments point-by-point, as well as the corresponding changes made to the revised manuscript. The reviewer's comments are marked in black and our answers are marked in blue, and the revision in the manuscript is further formatted as '*Italics*'.

**1. Summary**

The authors present measurements from Beijing, focusing on analysis and interpretation of data from a single particle soot photometer. The use the SP2 measurements to infer light absorption and light absorption amplification factors. The technical analysis is of reasonably high quality. The interpretation and discussion could benefit from some stronger quantitative analysis to discern process-based information. The authors also need to clarify how they have calculated averages, and whether they are mass-weighted or not. My specific comments follow below.

**Response:** We thank the reviewer for raising the important issue. In the revised manuscript, the average values of the $D_p/D_c$ ratio, the calculated $E_{ab}$, MAC, SFE and DRF were mass-weighted across all sizes above the detection limit of SP2 incandescence (i.e., rBC larger than ~75 nm). The mass-weighted averages were shown in the new Fig. 2b, Fig. 6, Fig. 7 and Table 1. The change rate of $E_{ab}$ shown in the Fig. 4 was also obtained based on the mass-weighted averages of the calculated $E_{ab}$.

**2. Specific comments**

(1) General comment on terminology: Throughout the manuscript, the authors need to

clarify when they are talking about actual absorption measurements or absorption enhancement measurement, and when they are talking about calculated/theoretical values. There are many, many points where this distinction needs to be made clearly, starting from the abstract and continuing through the conclusions. I will note only a few points where this is necessary as examples, but there are many more beyond what I have noted.

P1 L20 & 27, and P2 L1 & L3: This should be changed to "theoretical light absorption capability"

**Response:** Thanks for the suggestion. Throughout the manuscript, we have clarified the statement on actual absorption measurements and theoretical values.

(2) P2, L14: The Moffett study does not directly measure light absorption. Their conclusions are based on theoretical calculations. Thus, it is not appropriate here. Same for the Zhang et al. 2016 reference.

**Response:** Thanks and we agree with the reviewer. The references of Moffett et al. (2009) and Zhang et al. 2016 (2016) have been removed. We have added some other citations (Knox et al., 2009; Peng et al. 2016; Schnaiter et al. 2005), which reported the directly measure light absorption.

(3) P2, L15: It is unclear as written when the authors are referring to theoretical studies versus observations studies. This must be clarified.

**Response:** Thanks. Following the suggestion, the statement has been changed as "*Previous theoretical (Jacobson, 2001; Moffet et al., 2009; Zhang et al., 2016) and observation studies (Cappa et al., 2012; Peng et al., 2016; Knox et al., 2009) showed a broad range of absorption enhancements (1.05-3.05) of BC during the atmospheric aging process.*"

(4) P2, L23: This should be revised to clarify that the concept of "more coating

materials results in stronger light absorption capability" depends on whether one considers coatings on individual particles versus coatings averaged over the ensemble of particles.

**Response:** Thanks for the comments. To clarify it, we have revised the statement as *"In terms of individual BC particle, more coating materials results in its stronger light absorption capability."*

(5) P2 ,L29: It should be clarified that the idea that more materials coat BC under polluted conditions is only true so far as the total amount of BC does not scale equally with the overall amount of pollution. If there were more secondary aerosol but also more BC particles, then it is possible that the average coating per particle is unchanged in more versus less polluted conditions. Also, this statement oversimplifies issues related to mixing state, and whether that secondary material condenses on BC versus on non-BC containing particles. The authors oversimplify here, in my opinion.

**Response:** Thanks the reviewer to point this out. To clarify it, the sentence has been revised as *"Whether the changes of secondary aerosols with air pollution will affect the coating materials on the BC is complex, which not only depends on the increase in BC amount versus secondary aerosols but also controlled by secondary material condensation on BC versus non-BC containing particles."*

(6) P2, L31: it is unclear what "quasi-atmospheric" means. Also, this study ultimately indicates very simply that when you have greater amounts of coating on monodisperse particles the absorption enhancement is larger.

**Response:** Thanks. To make it clear, we have changed "quasi-atmospheric" into *"Recent BC aging measurements in Beijing and Houston using an environmental chamber (flowing ambient air to feed with lab-generated fresh BC particles)……"*

(7) P4, L3: Fig. S2 indicates that the uncertainty in the aethelometer measurements is

10% based on the uncertainty in the "compensation factors." However, this inherently assumes that the measurements by the reference instrument, a MAAP, are perfect and without uncertainty, which is not true. The actual uncertainty is larger and this should be noted. Also, this assumes that the MAAP perfectly accounts for filter-loading and multiple-scattering effects.

**Response:** Thanks for the comment and we agree with the reviewer. In this study, we also corrected the MAAP data using the algorithm reported by Hyvärinen et al. (2013). As the reviewer said, the uncertainty in the compensation factors of AE measurements obtained in our study depends on the uncertainty in MAAP measurements. Hyvärinen et al. (2013) compared the results from a PAS against those derived from the MAAP in Beijing, and estimated the uncertainty of ~15% in absorption coefficients derived from MAAP based on the developed algorithm. Therefore, we estimated that the factor *C* derived by comparing AE and MAAP measurement would exhibit an uncertainty of ~15%.

To make the statement more appropriate, the related statement in the supplementary has been revised as "*In this study, the uncertainty in the factor C was dominated by the uncertainty in MAAP measurements. We corrected the MAAP data using the algorithm reported by Hyvärinen et al. (2013). They estimated that the uncertainty in absorption coefficients derived from MAAP based on the developed algorithm was ~15% by comparing the results from a PAS against those derived from the MAAP in Beijing. This indicated that the factor C used in our study (~2.6) would exhibit an uncertainty of ~15% from the uncertainty in MAAP measurements. Considering the uncertainty on the AE33 measurements was mainly from the factor C, the absorption coefficient from AE33 was estimated to have an uncertainty of ~15%.*"

(8) Eqn. 1: This is not so much an equation as a relationship. It does not seem to me that it needs to be called out as an equation.

**Response:** Thanks. We have removed the relationship (1) and added the related

discussion *"In Mie calculation, the $D_p$ is retrieved from $C_s$, the size of rBC core ($D_c$) and the refractive indices of the non-BC shell ($RI_s$) and rBC core ($RI_c$)."*

(9) SP2 limits: The authors indicate a lower size limit of 75 nm in the main text. But, Fig. S4 makes clear that the lowest two size bins are strongly biased low in terms of their concentrations since there is no physical reason for such a sharp falloff in concentration below 95 nm. It is unclear whether this is taken into account, which would be particularly important when the [PM1] is < 50 ug/m3. Fig. S4 makes clear that there is a counting artifact in the SP below 95 nm, where the detection efficiency falls off rapidly giving rise to an apparent sharp decrease in concentration.

**Response:** Thanks and we agree with the reviewer. Considering the counting artifact in SP2 for smaller BC particles, the rBC mass concentration was corrected for SP2 detection efficiency. Figure R1 shows the SP2 detection efficiency concentration ($\eta$) in each rBC size-bin. In our study, the SP2 detection efficiency was determined with a DMA-SP2/CPC system. Monodispersed Aquadag particles generated by DMA were simultaneously measured by SP2 and CPC. The size-resolved $\eta$ was calculated by dividing the particle number concentration from SP2 measurement by that from CPC measurement.

[Figure]

Figure R1 (Fig. S3 in the revised manuscript). SP2 detection efficiency of particle (η) in each rBC size-bin.

To make this point clear, we have added the Fig. S3 (Fig. R1 in the response) and its related discussion in the revised manuscript: "*The mass concentration of rBC is calculated from the particle-to-particle mass of rBC and the sampled flow (~0.12 lpm). Note that the SP2 detection efficiency (Fig. S3) have been considered in the calculation of rBC mass concentration.*"

(10) Fig. S7 and P6/L5: it is unclear how this figure addresses uncertainties in the absorption calculations. However, the authors do argue in the supplemental that the "absorption: : :was underestimated by no more than 50%." A 50% uncertainty is very large and this information should not be buried in the supplemental. Further, additional details are required as to how this was determined. The MAC from Mie theory varies as a function of particle size while that from RDG is constant. The RDG MAC for bare BC, using the RI given here, is 3.2 m2/g at 880 nm. This is relatively small to begin with, so how is the 50% number determined. Also, the argument that "the uncertainty of BC light absorption from the calculation of bare BC properties using Mie theory is no more than 2%" is demonstrably not correct. If the absolute absorption can be underestimated by 50%, then the uncertainty cannot be only 2%. Just because there were few times that bare BC particles were observed does not change this fundamental issue. The uncertainty in the absolute absorption from the calculations is much larger than 2%. And then on P8, L5 it is stated that the uncertainty is 10%. It is ultimately unclear what the actual uncertainty on the calculations is.

**Response:** Thanks to the reviewer to point this out. We have recalculated the uncertainties on the calculated light absorption. In this study, the MAC for bare BC derived from Mie calculation, using the RI given here, is 3.8-4.5 $m^2$/g at 880 nm with

an average of ~4.3 m²/g (Fig. R2 in the response). Bond and Bergstrom (2006) suggested a value of 7.5 m²/g for the MAC of bare BC at 550 nm. Considering that the absorption is inversely proportional to wavelength (Bond and Bergstrom, 2006), the MAC of bare BC at 880 nm is estimated to be ~4.7 m²/g, which was slightly greater than that (~4.3 m²/g) obtained from Mie calculation in our study. This indicated the uncertainty of MAC for bare BC from Mie calculation was ~8%.

Correspondingly, we deleted the Fig. S7 and added the new Fig. S9 and related discussion in the revised supplement, as "*Based on Mie calculation, we obtained the MAC of rBC core (MAC$_c$) at 880 nm in the range of 3.8-4.5 m²/g with an average of ~4.3 m²/g during the campaign period (Fig. S9). Bond and Bergstrom (2006) suggested a value of 7.5 m²/g for the MAC of bare BC at 550 nm. Considering that the absorption is inversely proportional to wavelength (Bond and Bergstrom, 2006), the MAC of bare rBC at 880 nm is estimated to be ~4.7 m²/g, which was slightly greater than that (~4.3 m²/g) obtained from Mie calculation in our study. This indicated the uncertainty of MAC for bare rBC from Mie calculation was ~8%. We estimated that the uncertainties of calculated BC light absorption related to MAC of bare rBC from Mie calculation was ~8%.*"

[Figure]

Figure R2 (Fig. S9 in the revised manuscript). The time series of MAC derived from Mie calculation for BC cores (i.e., bare BC) at 880 nm.

(11) P7/L19: This should be Fig. S4a.

**Response:** We apologize for the typo and have revised it.

(12) P7/L26: This should say "calculated absorption coefficient."

**Response:** Thanks. We have revised it.

(13) P8/L2: As noted above, the uncertainty on the AE33 measurements is >10%.

**Response:** Thanks. As explained in the response to comment 7, we agree with the reviewer that the uncertainty on the AE33 measurements may larger than 10%. We estimated the uncertainty in factor $C$ was ~15% (see the response to comment 7). Considering the uncertainty on the AE33 measurements was mainly from the factor $C$, the absorption coefficient from AE33 was estimated to have an uncertainty of ~15%. To make the statement here more appropriate, we revised the discussion on the uncertainty in the AE33 measurements, see the response to comment 7.

(14) P8/L5: is the agreement similarly good at the other wavelengths?

**Response:** Thanks to the reviewer for raising this concern. We have compared the light absorption coefficient ($\sigma_{ab}$) at other wavelength (i.e., 660 nm) obtained from Mie calculation with that from AE33 measurement (Fig. R in the response). We found that the calculated $\sigma_{ab}$ and measured $\sigma_{ab}$ showed better correlation at a wavelength of 880 nm ($\sigma_{ab,calculated} = 0.90\sigma_{ab,measured}$ ($R^2$=0.98), shown in the Fig. 1c in the manuscript) than at a wavelength of 880 nm ($\sigma_{ab,calculated} = 0.74\sigma_{ab,measured}$ ($R^2$=0.97), shown in the Fig. R3 in the response). At the wavelength of 880 nm, other aerosol particles (carbonaceous or mineral) absorb significantly less and absorption can be attributed to BC alone. In this study, the calculated $\sigma_{ab}$ based on Mie theory characterized the absorption of rBC.

Therefore, we just compared the calculated $\sigma_{ab}$ at 880 nm with the measured $\sigma_{ab}$ using AE33 at 880 nm.

[Figure]

Figure R3 (Fig. S7 in the revised manuscript). The correlation between the calculated $\sigma_{ab}$ ($\sigma_{ab, calculated}$) at 660 nm using Mie theory combined with SP2 measurements and the measured $\sigma_{ab}$ ($\sigma_{ab, measured}$) by the AE33.

(15) P8/L9: It should be clarified that these are all theoretical studies of the enhancement. None of the citations is to a direct measurement of the enhancement.

**Response:** Thanks for the suggestion. We have revised the sentence as "*Previous theoretical studies reported that the coating materials on the BC surface can significantly enhance the light absorption of BC via the lens effect (Fuller et al., 1999; Jacobson, 2001; Lack and Cappa, 2010; Moffet et al., 2009).*"

(16) P8/L14: It should be clarified to "the calculated Eab".

**Response:** Thanks. Throughout the manuscript, we have revised it.

(17) P8/L16: It would be helpful to show a graph that explicitly has the Dp/Dc and

calculated Eabs as a function of PM1 concentration, to help illustrate this point. This could be shown as a mean across all sizes, a weighted mean across all sizes, or for a few select sizes. This would help to illustrate the magnitude of the changes.

**Response:** Following the reviewer's suggestion, we have added a graph that explicitly has the $D_p/D_c$ and calculated $E_{ab}$ as a function of PM1 concentration (Fig. R4 in the response and new Fig. 2b in the revised manuscript). The $D_p/D_c$ and calculated $E_{ab}$ values shown in the Fig. R3 are the mass-weighted averages across all sizes above the detection limit of SP2 incandescence (i.e., rBC larger than ~75 nm).

[Figure]

Figure R4 (Fig. 2b in the revised manuscript). Variations in the $D_p/D_c$ and calculated $E_{ab}$ of BC-containing particles with the PM$_1$ concentration. The $D_p/D_c$ and calculated $E_{ab}$ values shown in the Fig. 2b are the mass-weighted averages across all sizes above the detection limit of SP2 incandescence (i.e., rBC larger than ~75 nm).

Correspondingly, the related discussion on the new Fig. 2b in the revised manuscript has been added "*On average (i.e., mass-weighted mean across rBC core size larger than ~75 nm), the $D_p/D_c$ and calculated $E_{ab}$ for observed BC-containing particles in SP2 under different PM$_1$ concentrations during the campaign period varied in the range of 1.6-2.2 and 1.6-2.0, respectively (Fig. 2b). Correspondingly, the mass-averaged values of the $D_p/D_c$ and calculated $E_{ab}$ of BC-containing particles increased by ~33% and ~18%, respectively, with increasing PM$_1$ concentrations from*

*~10 μg m⁻³ to ~230 μg m⁻³.*"

(18) P8/L24: I find the term "aging degree" to be ambiguous because it could apply to almost anything. I suggest that here, and throughout, the authors change to a more specifically descriptive language. Perhaps "coating-to-core ratio"?

**Response:** Thanks for the suggestion. We have stated in the Methods section that the aging degree of BC-containing particles was characterized by the $D_p/D_c$ ratio in this study. Throughout the manuscript, we have changed some "aging degree" into "$D_p/D_c$ ratio".

(19) P8/L25: It would be helpful to frame this in the context of the overall size distribution, i.e. to report the weighted-average values based on the observed $PM_1$-dependent BC core size distributions.

**Response:** Thanks for the suggestion. We have added the new Fig. 2b in the revised manuscript (Fig. R4 in the response) to show the weighted-average values of $D_p/D_c$ and calculated $E_{ab}$ based on the observed PM1-dependent BC core size distributions. Please see the response to the comment 17.

(20) P8/L23: Better as "exponential function with a y-offset". However, this by itself gives little physical insight. In Metcalf et al. (2013), for example, there was a comparison to the expected decay based on the SA/V ratio of the particles and diffusion controlled growth. Here, the authors mention this study but do not connect to it quantitatively. The authors should strongly consider introducing a physical explanation using a semiquantitative analysis, rather than just an empirical fit. I believe this would strengthen the paper.

**Response:** This is a very good point. We have deleted the empirical fit. Following the reviewer's suggestion, we have introduced a physical explanation using a semiquantitative analysis (Fig. R5 in the response and new Fig. 3 in the revised

manuscript). The method of semiquantitative analysis shown here was similar with that using in Metcalf et al. (2013) and is described below.

[revised manuscript text omitted]

(21) P8/L30: it is unclear how the "change rates" were calculated. Are these point-by-point differences? And, how is the Eab calculated? Is this a weighted average? Also, it's not entirely demonstrated how this is an especially meaningful metric. Isn't the same general information obtained by plotting Eab vs. rBC (for example)? Assuming this is from point-by-point differences, then one would expect that:

k_Eab/k_pm1 =((E_(ab,t2)-E_(ab,t1))/E_(ab,t1) )/((PM1_t2-PM1_t1)/(PM_t1 ))=(PM1_t1)/E_(ab,t1) (E_(ab,t2)-E_(ab,t1))/(PM1_t2-PM1_t1 )

How is this generally useful? This could be elaborated upon. Also, given that negative values are allowed, it could be clarified that these are not just "growth" rates. This is really just a susceptibility curve.

**Response:** Thanks and yes. We determined the change rates from point-by-point differences, i.e., $k_{Eab} = (E_{ab,t2}-E_{ab,t1})/(E_{ab,t1}*(t2-t1))$, $k_{PM1} = (PM_{1,t2}-PM_{1,t1})/(PM_{1,t1}*(t2-t1))$ and $k_{rBC} = (C_{rBC,t2}-C_{rBC,t1})/(C_{rBC,t1}*(t2-t1))$. We have calculated the hourly $E_{ab}$ values with weighted average and used them to determine the change rates. The $k_{Eab}$, $k_{PM1}$ and $k_{rBC}$ values represent an apparent change rate of calculated $E_{ab}$, $PM_1$ concentration and rBC mass concentration, respectively. The $k_{Eab}/k_{PM1}$ characterizes the sensitivity of the change of calculated $E_{ab}$ with increasing/decreasing $PM_1$ concentrations. The same general information obtained by plotting $k_{Eab}$ vs. $k_{rBC}$ (i.e., the sensitivity of the change of calculated $E_{ab}$ with increasing/decreasing rBC mass concentration). We agree the reviewer that the $k_{Eab}$, $k_{PM1}$ and $k_{rBC}$ values shown in Fig. 3 (new Fig. 4 in the revised manuscript) are not just "growth" rates. We have changed the "growth rate" into "*change rate*" in the manuscript.

To make these points clear, we have added the statement on the "change rates" in the caption of new Fig. 4 in the revised manuscript, as "*The $k_{Eab}$, $k_{PM1}$ and $k_{rBC}$ values represent an apparent change rate of calculated $E_{ab}$, $PM_1$ concentration and rBC mass concentration, respectively, and are from point-by-point differences of hourly $E_{ab}$, namely $k_{Eab} = (E_{ab,t2}-E_{ab,t1})/(E_{ab,t1}*(t2-t1))$, $k_{PM1} = (PM_{1,t2}-PM_{1,t1})/(PM_{1,t1}*(t2-t1))$ and $k_{rBC} = (C_{rBC,t2}-C_{rBC,t1})/(C_{rBC,t1}*(t2-t1))$. The sensitivity of the change of calculated $E_{ab}$ with changing $PM_1$ and rBC concentrations was obtained by plotting $k_{Eab}$ vs. $k_{PM1}$ (i.e., $k_{Eab}/k_{PM1}$, the slope shown in (a)) and $k_{rBC}$ (i.e., $k_{Eab}/k_{rBC}$, the slope shown in (b)), respectively.*"

(22) P8/L31: The units on the equation are incorrect. It is keab = 4.8kPM1, not 4.8%. The percents cancel.

**Response:** Thanks. Figure 3a shows the linear relationship between $k_{Eab}$ and $k_{PM1}$ with a slope of 0.048, i.e. $k_{Eab}$=0.048$k_{PM1}$. To make this point clear, we have revised

the equation as "$k_{Eab}=0.048k_{PM1}$".

(23) P9/L1: The statement here relates to one point on a graph of hundreds of points. What is the uncertainty on a single point? Is this meaningful to state? I question whether it is especially meaningful to state the results for this one point. I could randomly pick another point, based on the maximum kPM1 (for example) and conclude that kEab varies slowly with kPM1. This feels to me too selective to be meaningful to include and I suggest it is removed or put in a fuller context.

**Response:** Thanks for the comment and we agree with the reviewer. Following reviewer's suggestion, this sentence has been removed from the revised manuscript.

(24) P9/L2: The authors relate their observations to other studies. However, I do not understand why they only consider values with kEab > 0. Why exclude the negative numbers? Also, to reiterate my above point, are the individual points truly meaningful once one accounts for the uncertainty in the individual points? Using confidence intervals for the slopes would, in my opinion, be more meaningful. Or looking at the distribution of kEab values. It is evident from Fig. 3 that if a histogram of kEab values was made the peak would be around zero, i.e. that the particles are shrinking as often as they are growing, on average. While I do see some value in providing the range of values here, a much more statistical picture would provide much greater value.

**Response:** Thanks. In the revised manuscript, we have considered $k_{Eab}$ values including both positive and negative numbers. Following reviewer's suggestion, we have removed the statement on the results based on the individual points and have added some statistical results (distributions of $k_{Eab}$, $k_{PM1}$, and $k_{Eab}/k_{PM1}$ values (Fig. R6 in the response and new Fig. 4b in the revised manuscript).

Correspondingly, the related discussion on the statistical results has been added in the revised manuscript "*Figure 4b shows frequency distribution of $k_{Eab}$, $k_{PM1}$, and $k_{Eab}/k_{PM1}$. During the campaign period, most of $k_{Eab}$ and $k_{PM1}$ values were in the range of -50%-50% $h^{-1}$ and -4%-4% $h^{-1}$, respectively, revealing a lower change rate for BC*

*aging than that for PM₁ concentration. The peak value of frequency distribution of*
*$k_{Eab}$ was around zero, indicating the BC particles are shrinking as often as they are*
*growing. The $k_{Eab}/k_{PM1}$ ratio characterized the sensitivity of the change of calculated*
*$E_{ab}$ with changing PM₁ concentrations. The frequency distribution of $k_{Eab}/k_{PM1}$ ratio*
*showed that ~60% values were in the range of 0-1, with a peak value around 0.05.*
*Smaller values of $k_{Eab}/k_{PM1}$ ratio indicated that the change of calculated $E_{ab}$ was not*
*sensitive to variations in PM₁ concentrations."*

[Figure]

Figure R6 (new Fig. 4 in the revised manuscript). (a) Correlation between the chaning rate of calculated $E_{ab}$ ($k_{Eab}$) and the chaning rates of PM₁ concentrations ($k_{PM1}$) during the campaign period. (b) Frequency distribution of $k_{Eab}$, $k_{PM1}$, and $k_{Eab}/k_{PM1}$. The $k_{Eab}$

and $k_{PM1}$ values represent an apparent change rate of $E_{ab}$, $PM_1$ concentration and rBC mass concentration, respectively, and are from point-by-point differences of hourly $E_{ab}$, namely $k_{Eab} = (E_{ab,t2}-E_{ab,t1})/(E_{ab,t1}*(t2-t1))$ and $k_{PM1} = (PM_{1,t2}-PM_{1,t1})/(PM_{1,t1}*(t2-t1))$. The sensitivity of the change of $E_{ab}$ with changing $PM_1$ concentrations was obtained by plotting $k_{Eab}$ vs. $k_{PM1}$ (i.e., $k_{Eab}/k_{PM1}$, the slope shown in (a)).

(25) Fig. S9 and P9/L5: If the authors were to introduce a more quantitative picture that included an interpretation of why this type of behavior might be expected that would be most welcome. Most likely, this is simply because the net change in diameter for a given amount of material deposited decreases with the size of the particle due to surface-to-volume scaling. Since the particles are larger when PM1 is larger, one would expect the deltaDp/time to decrease with PM1 and thus the Eab/time would also decrease. Of course, this oversimplifies because Eab is not a linear function of deltaDp. But, it would be great if the authors could introduce some physical discussion of why this observed behavior is/is not expected this would increase the value of this observation.

**Response:** Thanks. Following the reviewer's suggestion, we have introduced a more quantitative picture (Fig. R7 in the response and new Fig. S9b in the revised manuscript) and added some physical discussion on why the change rate of calculated $E_{ab}$ decreased with $PM_1$, as "*This can be explained by larger BC particles when $PM_1$ concentration is higher (Fig. S9b). The net change in diameter for a given amount of material deposited decreases with increasing particle size due to surface-to-volume scaling, which would expect the growth rate of particles to decrease with increasing $PM_1$ concentration and thus the $k_{Eab}$ would also decrease (Fig. S9a).*"

[Figure]

Figure R7 (new Fig. S9b in the revised manuscript). (b) Variations in the diameter of BC-containing particles ($D_p$) with the normalized $PM_1$ concentrations.

(26) P9/L9: I find this discussion about previous studies "ignoring" an aspect to be unclear and suggest it be expanded/clarified. How does the current observation help, specifically, explain these previous studies? It is not abundantly clear. Consider that the variability in the calculated Eabs is actually only 15% between the low and high PM1 periods in the size range that matters (the BC mass weighted size), based on Fig. 2. These previous measurements would not have been able to discern a 15% difference easily, most likely, in their data anyway. While the current study finds a large theoretical enhancement, what is not found is substantial variability in the enhancement (Fig. 2). The variability in the "growth rate" is inconsequential in the context of the actual enhancement dependence on PM1 (Fig.2). Related, it is not clear where the 28% on L19 comes from. What matters is the mass-weighted enhancement. Assuming this is a straight average over the points in Fig. 2, this is not relevant to the actual measurement of the enhancement, which is weighted. I suggest that the authors rethink this discussion entirely.

**Response:** Thanks for the comments. Following the reviewer's suggestion, we have rethink this discussion entirely and revised the statement in P9/L9-19, as "The evolution of theoretical light absorption of BC with pollution levels depends on the

change in both rBC mass concentrations and calculated $E_{ab}$. *Figure S13 shows markedly smaller $k_{Eab}$ than $k_{Rbc}$ ($k_{Eab} \approx 0.027\ k_{rBC}$), indicating the change of calculated $E_{ab}$ was significantly slower than that of rBC mass concentrations under different pollution levels. Due to less sensitive for calculated $E_{ab}$ to change in air pollution levels compared with that for rBC mass concentrations, some previous measurements (McMeeking et al., 2011; Ram et al., 2009; Wang et al., 2014b; Andreae, et al., 2008) would not have been able to discern a difference of $E_{ab}$ easily among different pollution levels and thus just focus on the change of BC mass concentration. This would lead to uncertainties in estimation of BC light absorption. In our case, we found the mass-weighted average of $E_{ab}$ increased by ~18% with $PM_1$ concentration increasing from 10 μg m$^{-3}$ to 230 μg m$^{-3}$ (Fig. 2b). If the change of calculated $E_{ab}$ of BC with $PM_1$ increase was neglected in our study, the theoretical light absorption of BC-containing particles would be underestimated by ~18% under polluted conditions.*"

(27) Related to the previous comment, it is unclear how the references on P9/L9 relate to the references on P9/L13. The authors appear to be linking these, I think, but it is not clear. Also, this seems selective, as there are studies (e.g. Liu et al. 2015) in which variability was observed. The authors should aim to provide a more comprehensive picture. Finally, it is not at all clear that the conditions in the cited studies are similar enough to those here to be relevant. This aspect needs to be discussed.

**Response:** Thanks to the reviewer for raising this concern. We have revised the statement in P9/L9-19, please see the comment (26).

(28) P9/L27: Should be Figure 4.

**Response:** We apologize for the typo and have revised it.

(29) Figure 4: The site location should be clearly indicated. In addition, the boundaries of the in-region vs. out of region should be indicated clearly. Also, it is not clear whether these regions are defined based on some physical parameter (e.g. as air

basins) or simply based on political boundaries. It would be useful if this were addressed.

**Response:** Thanks for the comment. We have indicated the site location and the boundaries of the in-region (i.e., Beijing) vs. out of region (i.e., other areas such as Tianjin, Heibei, Inner Mongolia, Shanxi, Shandong), shown in Fig. S1 (Fig. R8 in the response). These regions are defined based on political boundaries.

[Figure]

Figure R8 (Fig. S1 in the manuscript). Location of the observation site (red star).

To clarify it clear, we have added the statement in the caption of Fig. 5 in the revised manuscript, as "*The site location and the boundaries of the in-region (i.e., Beijing) vs. out of region (i.e., other areas such as Tianjin, Heibei, Inner Mongolia, Shanxi, Shandong), shown in Fig. S1. Noted that these regions are defined based on political boundaries.*"

(30) P10/L3: It is unclear where the 62% number comes from.

**Response:** Thanks for the comment. The percentage of 62% is obtained from the ratio between the 4.6-fold increase in $EEI_{total}$ and 7.4-fold increase in rBC mass concentration. Table 1 shows that rBC mass concentration increased by 7.4-fold from clean period (0.82 µg m$^{-3}$) to polluted period (6.07µg m$^{-3}$). The increase in rBC mass

concentration can be attributed to more BC transported to the site and the adverse local meteorology. In this study, the amount of BC transported to the site was characterized by $EEI_{total}$. Table 1 shows that nomalized $EEI_{total}$ increased by 4.6-fold from clean period (3.68) to polluted period (16.87), revealing that the increase in amount of BC transported to the site contributed 62% of increase in rBC mass concentration with air pollution development.

To make it clear, the sentence has been revise as "*Table 1 shows that the BC concentrations from the clean period to the polluted period increase by ~7.4 times. The increase of EEI$_{total}$ (~4.6 times) accounted for ~62% the increase in BC mass concentrations (~7.4 times).*"

(31) P10/L14: It is unclear how Fig. 4c indicates that there were "higher aging rates." Can this be clarified?

**Response:** Thanks. Fig. 4c in the manuscript shows that the BC particles transported to the site during polluted period were mainly from Hebei province, which is one of the polluted regions in China. Peng et al. (2015) pointed out higher BC aging rates under more polluted environments, indicating that BC particles passing though polluted regions would show higher aging rates during atmospheric transport compared to that from clean regions.

To clarify it, the statement has been revised as "*On the other hand, compared with the BC carried in the clean air mass from the northwest of Beijing during the clean period (Fig. 4a), the BC in the polluted air mass underwent regional transport from the region south of Beijing (i.e., Hebei, one of the most polluted provinces in China with high pollutant emission) during the polluted period. Peng et al. (2015) pointed out higher BC aging rates under more polluted environments, indicating that BC particles passing though polluted regions would show higher aging rates during atmospheric transport than that from clean regions.*"

(32) P10/L15: "observed" should be "calculated." Also, is this a weighted average? A straight average across size? This needs to be clarified. A weighted average is most

appropriate. This comment applies to everywhere in the manuscript that values for Eab or Dp/Dc are mentioned. What sort of averages are these? This needs to be clarified.

**Response:** Thanks for the comment. We have changed "observed" into "calculated". Here and elsewhere in the manuscript that values for calculated $E_{ab}$ or $D_p/D_c$ are mentioned, we have showed their mass-weighted average values.

To make it clear, the sentence has been revise as *"The mass-average value of calculated $E_{ab}$ for BC-containing particles observed at our site were ~1.66, ~1.81 and ~1.91 during the clean, slightly polluted and polluted periods, respectively (Table 1)...."*

(33) P10: I find the discussion with respect to O3 is somew20hat lacking in detail and nuance. While O3 is lower during the polluted events, the concentration of precursors may be higher and this would contribute to aging. Additionally, photochemical processing is not the only possible pathway. Have the authors considered to what extent NO3 oxidation at night might be important?

**Response:** Thanks for the reviewer for raising this concern. We agree with the reviewer that photochemical process is not the only possible pathway and other chemical processes may be important. However, in this study, we focused on the effect of regional transport on BC aging process. We just roughly discussed the chemical process during BC aging. The chemical process of BC aging under polluted environment in china is complex, which involved photochemical oxidation and heterogeneous chemical production (Zheng et al. 2015). We will investigate the chemical process of BC aging under polluted environment in future.

In the revised manuscript, we have toned down the related discussion on the chemical process of BC aging, as *"When PM$_1$ concentrations were higher than ~120 μg m$^{-3}$, O$_3$ concentrations decreased to ~2 ppb. Zheng et al. (2015) has demonstrated the weakened importance of photochemistry in the production and aging of secondary aerosols in Beijing under polluted conditions due to decrease of oxidant concentrations. This indicated that the photochemical processing in BC aging may be*

*weakened under higher polluted levels (i.e., $PM_1 > 120\mu g$ $m^{-3}$). Noted that photochemical processing is not the only possible pathway in BC aging process and other pathways were not discussed in this study. The local aging process of BC might be enhanced by other pathways.*"

(34) P10/L24: The meaning of "taking more EEItotal more BC" is unclear.

**Response:** Thanks. In this study, the amount of BC transported to the site was characterized by $EEI_{total}$ (calculated by Eq. (8) in the manuscript). Larger $EEI_{total}$ values at a certain time (Fig. 5a in the manuscript) revealed that more BC in the site was transported from regional origins.

   To make this point clear, the sentence has been revised as "*On the other hand, the changes in the amount of BC from regional transport was characterized by variation of $EEI_{total}$, which was used to evaluate the contributions of regional transport to BC aging.*"

(35) P11/L17: Is the MAC range given here the increase over the baseline or the actual MAC range at 550 nm? I find this unclear. Also, is this mass weighted? The MAC varies with particle size.

**Response:** Thanks for the comment. The MAC range of BC-containing particles given here is the calculated MAC range at 550 nm based on Mie theory. We have shown the mass-weighted MAC values in Fig. R9 in the response (new Fig. 7a in the revised manuscript). To make it clear, the statement has been revised as "*Figure 7 shows that with increasing pollution levels (i.e., $PM_1$ increasing from ~10 $\mu g$ $m^{-3}$ to ~230 $\mu g$ $m^{-3}$) during the campaign period, the mass-averaged values of calculated MAC at 550 nm for BC-containing particles increased from ~11 $m^2$ $g^{-1}$ to ~14 $m^2$ $g^{-1}$, which resulted in the SFE of BC-containing particles increasing from ~0.7 $m^2$ $g^{-1}$ $nm^{-1}$ to ~0.9 $m^2$ $g^{-1}$ $nm^{-1}$.*"

[Figure]

Figure R9 (new Fig. 7a in the revised manuscript). Variations in the *MAC* at 550 nm of BC-containing particles with the $PM_1$ concentrations.

(36) P11/L19: Are the DRF values given related to the total BC? It is surprising that the increase is so small, given that the BC concentration itself increased by a factor of 7 or so. This could be clarified.

**Response:** We thank the reviewer for raising the important issue. The DRF values given here was not related to the total BC. In this study, we focused on investigating the effect of BC light-absorption capability on DRF. Therefore, the increase in DRF of BC with increasing pollution levels shown in the Fig.7c (in the revised manuscript) just considered the change in light-absorption capability of BC. The DRF values for BC-containing particles at different pollution levels were obtained by scaling the average *DRF* (0.31 W m$^{-2}$) of externally mixed BC from various climate models (Bond et al. 2013) with a scaling factor of $E_{ab}$ under different $PM_1$ concentrations. To make it clear, we added the statement in the caption of Fig. 7 in the manuscript, as *"The DRF values for BC-containing particles at different pollution levels were obtained by scaling the average DRF (0.31 W m$^{-2}$, Table S1) of externally mixed BC from various climate models (Bond et al. 2013) with a scaling factor of $E_{ab}$ under different $PM_1$ concentrations. In order to point out the effect of BC light-absorption capability on DRF under different $PM_1$ concentrations, we did not consider the*

*changes of total BC amount for DRF calculation in Fig .7c.*"

Following the reviewer's suggestion, we also estimated the change in DRF of BC related to both the mass concentration and theoretical absorption capability of BC. The related discussion was added in the revised manuscript, as "*Fig. 7c shows the DRF of BC increased by ~15% during the polluted period compared with that during the clean period. Meanwhile, the BC mass concentration increased by ~7 times (Table 1). If assuming the DRF of BC during the clean period to be ~0.5 W m$^{-2}$ based on calculation shown in Fig. 7c, it would increase to ~4 W m$^{-2}$ under polluted conditions, taking the increase in both of the mass concentration and theoretical absorption capability of BC.*"

(37) P12/L5 and P12/L8: It is not clear to me that the authors have demonstrated that there is a "speeding up" or "acceleration" of the coating process in the more polluted air. In fact, they seem to be arguing that photochemical processing is slower, but that there is longer time. This would actually go against the idea that there is a speeding up. This should be revisited. Associated with this, it is not clear to me that Fig. 7 is necessarily correct. The EEI analysis indicates that the contribution from the regional sources is smaller during less polluted periods. This does not mean that those particles from regional sources are less coated just because the overall particle distribution has less coating during low pollution periods. In fact, it is possible that the regional particles are more coated due to higher photochemical activity (potentially). But, because their fractional contribution is smaller the net impact on the coating amount appears smaller in the average, which is now dominated by the local sources. I think that Fig. 7 and the discussion section need to be rethought a little bit to provide a more nuanced picture of what might be happening. It may be that the authors are correct, but I do not think that they have fully justified their conclusion here.

**Response:** Thanks for the reviewer for raising this concern. Considering more coating precursors (e.g, $SO_2$, $NO_x$ and VOCs) in more polluted air mass, we pointed out a "speeding up" or "acceleration" of the coating process during atmospheric transport for regional BC particles in the polluted air compared with that in the clean air. Peng

et al. (2016) has revealed that compared with clean urban environments (i.e, Houston), the more efficient BC growth under polluted urban environments (i.e, Beijing) was attributable to higher concentrations of gaseous aerosol precursors.

It is noted that a "speeding up" or "acceleration" of the coating process mentioned here was just for regional BC. Under polluted period, the BC particles transported to the site was dominated by polluted regions (i.e., high emission areas such as south of Heibei province and Tianjin). However, under clean period, the BC particles transported to the site was dominated by clean regions (i.e., low emission areas such as Inner Mongolia). Polluted air mass from polluted regions exhibit more coating precursors. For regional BC particles, we evaluated their chemical activities (i.e., photochemical and other processes) based on the amount of coating precursors in air mass. We cannot separately estimate the photochemical and other processes during regional transport. The photochemical process under polluted conditions may be weakened during regional transport. However, photochemical process is not the only possible pathway and other chemical processes may be important. Zheng et al., (2015) has demonstrated the importance of both regional transport and heterogeneous chemistry in secondary aerosol production. On the other hand, although we discussed the photochemical production based on $O_3$ concentration in this study, this discussion just focused on local BC particles in Beijing.

Following the reviewer's suggestion, we have revised the Fig. 7 (Fig. R10 in the response) and related discussion to provide a more nuanced picture of what might be happening, as "*As shown in Fig. 8, this amplification effect on BC light absorption associated with air pollution is caused by increasing BC concentration and at the same time enhanced light absorption capacity of BC-containing particles by more coating production in the more polluted air. Variation of both the mass concentration and light absorption capability of BC associated with air pollution strongly depend on the air pollutant emission (e.g., BC, $SO_2$, $NO_x$ and VOC). Under polluted environment, polluted air mass from high emission areas not only brings more BC, but also more coating materials on BC surface due to more precursors of secondary components.*"

[Figure]

Figure R10 (new Fig. 8 in the revised manuscript). Conceptual scheme of amplification effect on BC light absorption associated with air pollution.

(38) P12, conclusions: The authors should consider reporting mass-weighted averages of Dp/Dc and Eab in addition to the ranges to provide a fuller picture.

**Response:** Thanks for the comment. Following the reviewer's suggestion, we have reported the mass-weighted averages of the $D_p/D_c$ ratio and calculated $E_{ab}$ values Correspondingly, the related statement was added in conclusions section, as "*During the campaign period, the hourly values of mass-weighted averages of the $D_p/D_c$ ratio and calculated $E_{ab}$ for BC-containing particles was in the range of 1.5-2.3 and 1.5-2.0, respectively. When PM$_1$ concentration increased from ~10 μg m$^{-3}$ to ~230 μg m$^{-3}$, the mass-weighted averages of the $D_p/D_c$ ratio and calculated $E_{ab}$ values increased by ~33% and ~18%, respectively.*"

(39) P13/L6: See previous comment regarding the reporting of single points without stated uncertainties. Is the 7.3%/h value believable? It is unclear, since it is a single outlier in the entire plot.

**Response:** Thanks. As explained in our response to comments (23) and (24), we have we have removed the statement on the results based on the individual points and have added the statement on some statistical results. In the conclusions section, the revised statement was "*During the campaign period, $k_{Eab}$ values were in the range of -4%-4% $h^{-1}$, with a peak of frequency distribution around zero, indicating that the BC particles are shrinking as often as they are growing. The frequency distribution of $k_{Eab}/k_{PM1}$ ratio showed that a peak value around 0.05, revealing that the change of calculated $E_{ab}$ was not sensitive to variations in $PM_1$ concentrations.*"

(40) P13/L10: It is unclear how a 13-44% variation in the Eabs translates to a 28% underestimate in absorption. This appears to simply be an average of 13% and 44%, and not an appropriately mass-weighted average.

**Response:** Thanks for the reviewer to point this out. As explained in our response to comment (26), we have recalculated the underestimate in absorption based on the changes in the mass-weighted average value of calculated $E_{ab}$. The related discussion was revised in conclusions section as "*In our case, if we had not considered the increase in the BC light absorption capability with increasing air pollution during the campaign period, the theoretical light absorption of BC-containing particles under polluted conditions would have been underestimated by ~18%.*"

(41) P13/L16: See previous comments about "speeding up". I do not think the authors have justified this conclusion.

**Response:** Thanks. As explained in our response to comment (37), the sentence has been revised as "*Not only more BC but also more coatings are carried into Beijing by more polluted regional air mass (Fig. 7 (a)), which can be explained by more coating precursors (e.g. $SO_2$, $NO_x$ and VOC) in a more polluted air.*"

(42) P13/L23: It is unclear where the conclusion regarding heterogeneous chemistry comes from. This is pure speculation that is introduced at this point without justification. Why do the authors believe this to be the case? Also, this seems arbitrary.

If the authors had defined their periods differently then they could come to a different conclusion.

**Response:** Thanks for the comment. We have deleted the statement on "heterogeneous chemistry" and the sentence has been revised as "*The further increase of $D_p/D_c$ (~2.0 to ~2.2) and $E_{ab}$ (~1.9 to ~2.0) associated with air pollution is harder and is mostly likely attributed to local chemical production.*"